# MALT Diffusion: Memory-Augmented Latent Transformers for Long Video Generation

## Abstract

Diffusion models are successful for synthesizing high quality videos but are limited to generating short clips (*e.g.* 2-10 seconds). Synthesizing sustained footage (*e.g.* over minutes) still remains an open research question. In this paper, we propose *MALT Diffusion* (using Memory-Augmented Latent Transformers), a new diffusion model specialized for long video generation. MALT Diffusion (or just MALT) handles long videos by subdividing them into short segments and doing segment-level autoregressive generation. To achieve this, we first propose recurrent attention layers that encode multiple segments into a compact memory latent vector; by maintaining this memory vector over time, MALT is able to condition on it and continuously generate new footage based on a long temporal context. We also present several training techniques that enable the model to generate frames over a long horizon with consistent quality and minimal degradation. We validate the effectiveness of MALT through experiments on long video benchmarks. We first perform extensive analysis of MALT in long-contextual understanding capability and stability using popular long video benchmarks. For example, MALT achieves an FVD score of 220.4 on 128-frame video generation on UCF-101, outperforming the previous state-of-the-art of 648.4. Finally, we explore MALT 's capabilities in a text-to-video generation setting and show that it can produce long videos compared with recent techniques for long text-to-video generation.

## 1 Introduction

Diffusion models (DMs) (Ho et al., 2020; Song et al., 2021b) provide a scalable approach to training high quality generative models and have been shown in recent years to apply widely across a variety of data domains including images (Dhariwal & Nichol, 2021; Karras et al., 2022), audio (Kong et al., 2021; Lakhotia et al., 2021), 3D shapes (Lee et al., 2024; Luo & Hu, 2021; Zeng et al., 2022), and more. Inspired by these successes, a number of DM-based approaches have also been proposed for generating videos (Blattmann et al., 2023; Gupta et al., 2023; Ho et al., 2022b; Bar-Tal et al., 2024) and even enabling complex zero-shot text-to-video generation (Gupta et al., 2023; Brooks et al., 2024).

In video generation, a central challenge and key desiderata is to achieve *any-length* generation where the model is capable of producing a video of arbitrary length conditioned on its understanding of the previous context. Many state-of-the-art existing video DM frameworks are *prima facie*, only capable of generating videos of fixed temporal length (*e.g.*, in the form of a fixed shape 3D RGB tensor (Ho et al., 2022b)) and generally limited from 2-10 seconds. A standard "naive" approach for addressing any-length video generation is to use autoregressive generation where new frames are generated conditioned on a small number of previously generated video frames (rather than a long context). For example, given a 4-second base model, one might autoregressively generate 2 new seconds of footage at a time, conditioning on the previous 2 seconds. Thus, this autoregressive approach can generate compelling videos for longer extents than the base model (especially for very high-quality generation models). However, it also has clear limitations. By not conditioning on long temporal extents, these models often struggle to achieve long-term consistency. Moreover, small errors during generation can easily cause error propagation (Ruhe et al., 2024), resulting in a rapid quality drop with the frame index.

To summarize, there are two key subchallenges entailed by the long video generation problem:

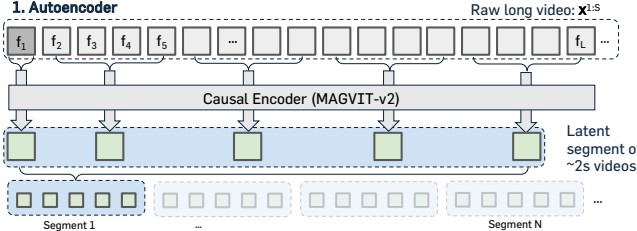 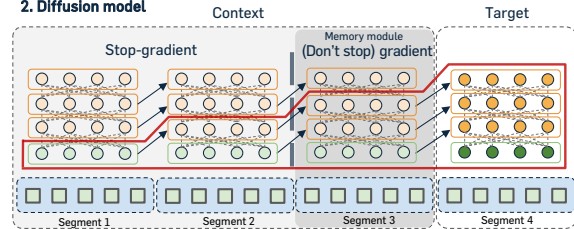

Figure 1: **Overview.** MALT first encodes a long video into a low-dimensional latent vector using a video autoencoder (*e.g.*, MAGVIT-v2 (Yu et al., 2024a)) and then divides it into multiple segments. After that, we recurrently generate these segments one-by-one, based on our newly proposed diffusion transformer architecture by adding recurrent temporal attention layers to existing architectures (Gupta et al., 2023; Peebles & Xie, 2023). This temporal attention layer augments the current keys and values by appending the hidden states from the previous computation to the keys. The red line on right figure denotes the increased receptive field of the MALT relative to a single segment.

- **long-term contextual understanding**: allowing the model to condition on long temporal extents while staying within memory limitations and computational constraints
- **long-term stability**: generating long videos without frame quality degradation over time.

Solving both of these issues is a critical next step for video generation, and indeed, there have been a number of recent works that generate stable video using long context (Harvey et al., 2022; Yan et al., 2023). However, they have been limited to simple datasets and fail to be stable for complex, realistic datasets.

**Our approach.** In this paper, we tackle any-length video generation by achieving the aforementioned twin goals of long-term contextual understanding and long-term stability. Toward this end, we introduce a new latent DM specialized for long videos, coined *MALT Diffusion*: Memory-Augmented Latent Transformers (MALT for short); see Figure 1 for an illustration. For long-term contextual understanding, we introduce a new diffusion transformer architecture that enables blockwise autoregressive generation of long videos. Motivated by recent language models developed for handling long sequences (*e.g.*, (Dai et al., 2019)), we design our model by adding recurrent temporal attention layers into an existing diffusion transformer architecture, which computes the attention between the current segment and the hidden states of previous segments (context). Hence, hidden states of contextual segments act as a "memory" to encode context and is used to generate future segments. Consequently, in contrast to conventional DMs, our model can behave as both a DM for generating a video clip and as an encoder to encode previous contexts (clean inputs) as memory.

To mitigate error accumulation at inference (*i.e.*, to achieve long-term stability), we propose a training technique that encourages our trained model to be robust with respect to noisy memory vectors by applying noise augmentation to the memory vector at training time. Moreover, we carefully design our model optimization to mitigate the rapid increase of computation and memory costs with respect to video length, enabling the training to be done on very long videos. With these components, MALT outperforms previous state-of-the-art results on popular long video generation and prediction benchmarks (Yu et al., 2023b; Yan et al., 2023) while using about 2× fewer parameters than the prior best baselines.[1]

The main contributions of our paper are:

- We propose a novel latent diffusion model *MALT Diffusion*: Memory-Augmented Latent Transformers (or just MALT), which can generate and be trained on long videos and addresses the aforementioned shortcomings.
- We propose several techniques for training our architecture on long videos, resulting in a robust model both capable of being a denoiser and an encoder of previous contexts. We also ensure training can be performed without expensive memory requirements.

---

[1]We mainly focus on videos, but ideas can be similarly applied to other domain like PDE or climate data (Ruhe et al., 2024).

- We show the strengths of MALT on long video generation/prediction benchmarks. For instance, MALT achieves an FVD (Unterthiner et al., 2018) of 220.4 in unconditional 128-frame video generation on UCF-101 (Soomro et al., 2012), 66.0% better than the previous state-of-the-art of 648.4. Also, on a challenging realistic dataset, Kinetics-600 (Kay et al., 2017), MALT exceeds the prior best long video prediction FVD as 799→392.

- We also qualitatively validate MALT's capability in long-term contextual understanding and stability. In particular, we show that MALT shows a strong length generalization: MALT generates long videos when trained on an open-world short text-to-video dataset (up to 4-5 seconds per clip), producing >120 seconds at 8 fps without suffering from significant frame quality degradation.

## 2 MALT Diffusion with Memory-Augmented Latent Transformers

Consider a dataset $\mathcal{D}$, where each example $(\mathbf{x}, \mathbf{c}) \in \mathcal{D}$ consists of a video $\mathbf{x}$ and corresponding conditions $\mathbf{c}$ (*e.g.*, text captions). Our goal is to train a model using $\mathcal{D}$ to learn a model distribution $p_{\text{model}}(\mathbf{x}|\mathbf{c})$ that matches a ground-truth conditional distribution $p_{\text{data}}(\mathbf{x}|\mathbf{c})$. In particular, we are interested in the situation where each $\mathbf{x}$ is a *long* video, and video lengths are much larger than conventional methods that often use $\sim$20-128 frames for both training and inference (He et al., 2022).

To efficiently model long video distribution, we adopt a compact "memory" that encodes previous long context in our latent diffusion transformer. Specifically, we aim to train a single model capable of: (a) encoding previous context of the long video as a compact memory latent vector, and (b) generating a future clip conditioned on the memory and $\mathbf{c}$.

In the rest of this section, we explain MALT Diffusion in detail. In Section 2.1, we provide a brief overview of latent diffusion models. In Section 2.2, we describe problem formulation and how we design a training objective. Finally, in Section 2.3, we explain the architecture that we used.

**Notation.** We write a sequence of tensors $[\mathbf{x}^a \ldots, \mathbf{x}^b]$ with $a < b$ as $\mathbf{x}^{a:b}$.

### 2.1 Latent diffusion models

In order to generate data, diffusion models learn the *reverse* process of a forward diffusion, where the forward diffusion diffuses an example $\mathbf{x}_0 \sim p_{\text{data}}(\mathbf{x})$ to a (simple) prior distribution $\mathbf{x}_T \sim \mathcal{N}(\mathbf{0}, \sigma_{\text{max}}^2 \mathbf{I})$ (with pre-defined $\sigma_{\text{max}} > 0$) with the following stochastic differential equation (SDE):

$$d\mathbf{x} = \mathbf{f}(\mathbf{x}, t)dt + g(t)d\mathbf{w}, \tag{1}$$

where $\mathbf{f}$, $g$, $\mathbf{w}$ are pre-defined drift and diffusion coefficients, and standard Wiener process (respectively) with $t \in [0, T]$ with $T > 0$. With this forward process, data sampling can be done with the following reverse SDE of Eq. equation 1:

$$d\mathbf{x} = \left[\mathbf{f}(\mathbf{x}, t) - \frac{1}{2}g(t)^2 \nabla_\mathbf{x} \log p_t(\mathbf{x})\right]dt + g(t)d\bar{\mathbf{w}}, \tag{2}$$

where $\bar{\mathbf{w}}$ is a standard reverse-time Wiener process, and $\nabla_\mathbf{x} \log p_t(\mathbf{x})$ is a score function of the marginal density from Eq. equation 1 at time $t$. Song et al. (2021b) shows there exists a *probability flow ordinary differential equation (PF ODE)* whose marginal $p_t(\mathbf{x})$ is identical to the reverse SDE:

$$d\mathbf{x} = \left[\mathbf{f}(\mathbf{x}, t) - \frac{1}{2}g(t)^2 \nabla_\mathbf{x} \log p_t(\mathbf{x})\right]dt. \tag{3}$$

Following previous diffusion model methods (Lee et al., 2024; Zheng et al., 2024), we use the setup in EDM (Karras et al., 2022) with $\mathbf{f}(\mathbf{x}, t) := \mathbf{0}$, $g(t) := \sqrt{2\dot{\sigma}(t)\sigma(t)}$ and a decreasing noise schedule $\sigma : [0, T] \to \mathbb{R}_+$, *e.g.*, $\sigma(t) = t$ and $\dot{\sigma}(t) = 1$. In this case, the PF ODE in Eq. equation 3 can be written:

$$d\mathbf{x} = -\dot{\sigma}(t)\sigma(t)\nabla_\mathbf{x} \log p(\mathbf{x}; \sigma(t))dt, \tag{4}$$

where $p(\mathbf{x}; \sigma)$ is the smoothed distribution by adding i.i.d Gaussian noise $\boldsymbol{\epsilon} \sim \mathcal{N}(\mathbf{0}, \sigma^2 \mathbf{I})$ with standard deviation $\sigma > 0$. To learn the score function $\nabla_{\mathbf{x}} \log p(\mathbf{x}; \sigma(t))$, we train a denoising network $D_{\boldsymbol{\theta}}(\mathbf{x}, t)$ with the denoising score matching (DSM) (Song & Ermon, 2019) objective for all $t \in [0, T]$:

$$\mathbb{E}_{\mathbf{x}, \boldsymbol{\epsilon}, t}\Big[\lambda(t) \|D_{\boldsymbol{\theta}}(\mathbf{x}_0 + \boldsymbol{\epsilon}, t) - \mathbf{x}_0\|_2^2\Big],$$

where $\lambda(\cdot)$ assigns a non-negative weight. and For simplicity, we uniformly sample $t$ and use $\lambda(t) = 1$.

However, training $D_{\boldsymbol{\theta}}$ directly with raw high-dimensional $\mathbf{x}$ is computation and memory expensive. To solve this problem, latent diffusion models (Rombach et al., 2022) first learn a lower dimensional latent representation of $\mathbf{x}$ by training an autoencoder (with encoder $F(\mathbf{x}) = \mathbf{z}$ and decoder $G(\mathbf{z}) = \mathbf{x}$) to reconstruct $\mathbf{x}$ from the low-dimensional vector $\mathbf{z}$, and then train $D_{\boldsymbol{\theta}}$ to generate in this latent space instead. Specifically, latent diffusion models use the following denoising objective defined in the latent space:

$$\mathbb{E}_{\mathbf{x}, \boldsymbol{\epsilon}, t}\Big[\lambda(t) \|D_{\boldsymbol{\theta}}(\mathbf{z}_0 + \boldsymbol{\epsilon}, t) - \mathbf{z}_0\|_2^2\Big],$$

where $\mathbf{z}_0 = F(\mathbf{x}_0)$. After training the model in latent space, we sample a latent vector $\mathbf{z}$ through an ODE or SDE solver (Song et al., 2021a;b; Karras et al., 2022) starting from random noises and then decode the result using $G$ to generate a final sample.

## 2.2 Modeling long videos via blockwise diffusion

Given a "long" video $\mathbf{x}^{1:S} \in \mathbb{R}^{S \times H \times W \times 3}$ with a resolution $H \times W$ and number of frames $S > 0$, we divide the video into $N$ *segments* of length $L$: $\mathbf{x}^{(i-1)L+1:iL}$ for $1 \le i \le N$ (thus $S = NL$). MALT will autoregressively generate these segments one-by-one.

**Autoencoder.** Following the standard latent diffusion approach, we would like to be able to encode and decode videos using a trained autoencoder, which is typically more lightweight compared to the diffusion model. However for very long videos, even running the autoencoder can be infeasible—thus we encode and decode videos in *chunks* of $m < N$ contiguous segments at a time. Specifically, for $1 \le i \le N/m$, we encode $\mathbf{x}^{(i-1)mL+1:imL}$ as a latent vector $\mathbf{z}^{im:(i+1)m}$ and decode it as:

$$\mathbf{z}^{im:(i+1)m} := F(\mathbf{x}^{(i-1)mL+1:imL}) \in \mathbb{R}^{m \cdot l \times h \times w \times c},$$

$$G(\mathbf{z}^{im:(i+1)m}) \approx \mathbf{x}^{(i-1)mL+1:imL},$$

where $F(\cdot)$ is an encoder network that maps the original video segments to their corresponding latent vectors with a spatial downsampling factor $d_s = H/h = W/w > 1$ and a temporal downsampling factor $d_l = L/l > 1$, and $G(\cdot)$ is a decoder network. We use these latent segments $\mathbf{z}^1, \ldots, \mathbf{z}^n$ of the original $\mathbf{x}$ obtained from the autoencoder for modeling the long video distribution.

**Diffusion model.** We now directly model the joint distribution of latent segments, $p(\mathbf{z}^{1:N}|\mathbf{c})$, autoregressively as

$$p(\mathbf{z}^{1:N}|\mathbf{c}) = \prod_{n=0}^{N-1} p(\mathbf{z}^{n+1}|\mathbf{z}^{1:n}, \mathbf{c}) \text{with } \mathbf{z}^{1:0} := \mathbf{0} \tag{5}$$

where we learn all $p(\mathbf{z}^{n+1}|\mathbf{z}^{1:n}, \mathbf{c})$ for $0 \le n \le N-1$ using a single diffusion model $D_{\boldsymbol{\theta}}$.

A naïve approach would be to use $\mathbf{z}^{1:n}$ directly as a condition to the model; however, if the number of segments, $N$, is large, $\mathbf{z}^{1:n}$ easily becomes extremely high-dimensional and prohibitive with respect to memory and compute. To mitigate this bottleneck, we instead introduce a *fixed-size* hidden state $\mathbf{h}^i := [\mathbf{h}_1^i, \ldots, \mathbf{h}_d^i]$ recurrently computed from $D_{\boldsymbol{\theta}}$. This hidden state serves as a memory vector to encode the context (i.e., the previous sequence of segments) $\mathbf{z}^{1:n}$, where $d > 0$ is a number of hidden states that are used as memory vectors (*i.e.*, $i$ is a segment index and $d$ refers to the number of layers of the model).

Specifically, for $1 \le i \le n$, we compute $\mathbf{h}^n$ with the following recurrent mechanism:

$$\mathbf{h}^i = \text{HiddenState}\big(D_{\boldsymbol{\theta}}(\mathbf{z}_0^i, 0; \mathbf{sg}(\mathbf{h}^{i-1}), \mathbf{c})\big), \tag{6}$$

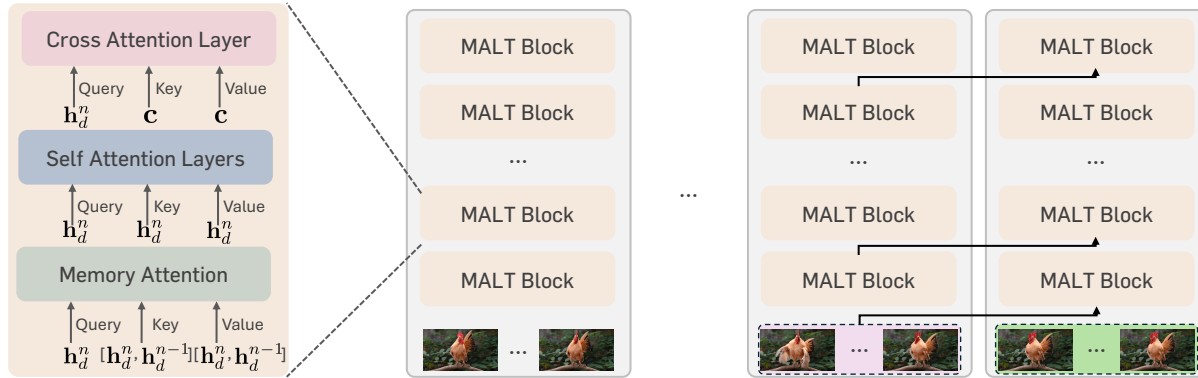

Figure 2: **MALT model architecture.** We add memory attention layers to the existing diffusion transformer architectures, which are operated recurrently across segments of short video clips.

where $\mathbf{h}^0 = [\mathbf{0}, \ldots, \mathbf{0}]$ and $\mathtt{sg}$ denotes a stop-grad operation. Note that we use a clean segment $\mathbf{z}^i$ without added noise so we set $t = 0$ here, which has not been used in conventional diffusion model training (Ho et al., 2020; Song et al., 2021a;b).

To sum up, we train $D_{\boldsymbol{\theta}}$ with the following denoising autoencoder objective with the memory $\mathbf{h}^n$:

$$\mathbb{E}\Big[\lambda(t)||D_{\boldsymbol{\theta}}(\mathbf{z}_t^{n+1}, t; \mathbf{h}^n, \mathbf{c}) - \mathbf{z}_0^{n+1}||_2^2\Big], \tag{7}$$

$$\mathbf{z}_t^{n+1} := \mathbf{z}_0^{n+1} + \boldsymbol{\epsilon}, \text{ where } \boldsymbol{\epsilon} \sim \mathcal{N}(\mathbf{0}, \sigma^2\mathbf{I}), \tag{8}$$

where each data is sampled from the video dataset $\mathcal{D}$, $t \sim [1, T]$, and $n \sim P(n)$ with a pre-defined prior distribution $P(n)$. Specifically, we set $P(n)$ as $P(n = 0) = 1/2$ and $P(n = k) = 1/2(N-1)$ for $k > 0$, as generating videos without memory (i.e., $n = 0$) is more difficult than continuation with a given memory vector (i.e., $n > 0$). Note that in Eq. 8 we do not use the stop-grad operation on $\mathbf{h}^n$ itself; as we mentioned, diffusion model training uses $t \geq 1$ in the common diffusion model objective because they only consider noisy inputs but our memory vector computation uses a clean sample ($t = 0$), which cannot be optimized without a backpropagation to $\mathbf{h}^n$. Instead, we use the stop-grad operation on the $\mathbf{h}^i$ for $i < n$ which are used to compute $\mathbf{h}^n$ in order to reduce memory requirements with respect to the number of segments used during training.

**Training for long term stability.** Unfortunately, the video quality from the model $D_{\boldsymbol{\theta}}$ trained with Eq. equation 8 is usually not satisfactory, because of frame quality degradation caused by error accumulation. We hypothesize that the reason why this happens is because there exists a discrepancy between training and inference: in training, we use ground-truth latent vector $\mathbf{z}^n$ for computation of the memory $\mathbf{h}^n$, but at inference, the model instead uses generated latent vectors, where small errors during generation can compound over a long sequence of segments.

To mitigate this discrepancy, we use a noisy version of $\mathbf{h}^n$ at training time, denoted by $\tilde{\mathbf{h}}^n$, where

$$\tilde{\mathbf{h}}^n := \mathrm{HiddenState}\big(D_{\boldsymbol{\theta}}(\mathbf{z}^n + \boldsymbol{\xi}, 0; \mathtt{sg}(\mathbf{h}^{n-1}), \mathbf{c})\big), \tag{9}$$

$$\text{where } \mathbf{h}^0 = [\mathbf{0}, \ldots, \mathbf{0}], \tag{10}$$

and $\boldsymbol{\xi} \sim p(\boldsymbol{\xi})$ with a pre-defined prior distribution $p(\boldsymbol{\xi})$. Since $\tilde{\mathbf{h}}^n$ is computed with $\mathbf{h}^{n-1}$ and a *noisy* latent vector $\mathbf{z}^n + \boldsymbol{\xi}$, the model is trained to be robust to small errors and reduces the train-test discrepancy between the memory computed at training and inference.

Summing up all of these components, our final training objective $\mathcal{L}(\boldsymbol{\theta})$ becomes:

$$\mathcal{L}(\boldsymbol{\theta}) := \mathbb{E}\Big[\lambda(t)||D_{\boldsymbol{\theta}}(\mathbf{z}_t^{n+1}, t; \tilde{\mathbf{h}}^n, \mathbf{c}) - \mathbf{z}_0^{n+1}||_2^2\Big], \tag{11}$$

with prior distributions $p(n), p(\boldsymbol{\xi})$.

For $p(\boldsymbol{\epsilon})$, we use a progressively correlated Gaussian distribution proposed in Ge et al. (2023) to further mitigate error accumulation. Next, we set $p(\boldsymbol{\xi})$ to be a Gaussian $\mathcal{N}(\mathbf{0}, \sigma_{\mathrm{mem}}^2 \mathbf{I})$ with small $\sigma_{\mathrm{mem}} > 0$.

**Inference.** After training, we synthesize a long video by autoregressively generating one segment at a time. Specifically, we start from generating a first segment $\mathbf{z}^1$ conditioned on $\mathbf{c}$, and then iteratively generate $\mathbf{z}^{n+1}$ for $n > 0$ by computing memory $\mathbf{h}^n$ and performing conditional generation from $\mathbf{h}^n$ and $\mathbf{c}$. We provide detailed pseudocode in Appendix A.

### 2.3 Architecture

**Autoencoder.** Similar to the encoding scheme used in a recent latent video diffusion model, W.A.L.T (Gupta et al., 2023), we use a causal 3D CNN encoder-decoder architecture for the video autoencoder based on the MAGVIT-2 tokenizer (Yu et al., 2024a) without quantization (so that latent vectors lie in a continuous space). We train the autoencoder with a sum of pixel-level reconstruction loss (*e.g.*, mean-squared error), perceptual loss (*e.g.*, LPIPS (Zhang et al., 2018)), and adversarial loss (Goodfellow et al., 2014) similar to prior image and video generation methods. Recall that both training and inference are not done directly on long videos; they are done after splitting long videos into short segments.

**Diffusion model.** As outlined in Figure 2, our model architecture is based on the recent diffusion transformer (DiT) architecture (Peebles & Xie, 2023; Ma et al., 2024b; Yu et al., 2024b). Thus, given a latent vector $\mathbf{z}^n \in \mathbb{R}^{l \times h \times w \times c}$ of a video clip $\mathbf{x}^n$, we patchify it with patch size $p_l \times p_s \times p_s$ to form a flattened latent vector $\texttt{patchify}(\mathbf{z}^n) \in \mathbb{R}^{(lhw/p_l p_s^2) \times c}$ with a sequence length $lhw/p_l p_s^2$ and use it as inputs to the model. In particular, we choose W.A.L.T (Gupta et al., 2023) as backbone, a variant of DiT which employs efficient spatiotemporal windowed attention instead of full attention between large numbers of video patches.

To enable training with long videos with DiT architectures, we introduce a memory-augmented attention layer and insert this layer to the beginning of every Transformer block. Specifically, we design this layer as a cross-attention layer between the previous memory latent vector and the current hidden state, similar to memory-augmented attention (Dai et al., 2019) in the language domain. Hence, query, key, and value of a $d$-th memory layer with the segment $\mathbf{z}^n$ and the memory latent vector $\mathbf{h}^{n-1}$ become:

$$\text{query} \coloneqq \mathbf{h}_d^n, \ \text{key} \coloneqq [\mathbf{h}_d^{n-1}, \mathbf{h}_d^n], \ \text{value} \coloneqq [\mathbf{h}_d^{n-1}, \mathbf{h}_d^n],$$
$$\mathbf{h}_d^{n-1}, \mathbf{h}_d^n \in \mathbb{R}^{(hw/p_s^2) \times (l/p_l) \times c'},$$

where $c'$ denotes the hidden dimension of the model and $\mathbf{h}_d^{n-1}, \mathbf{h}_d^n$ are *reshaped* as a sequence length $l/p_l$ and a batch dimension size $hw/p_s^2$, similar to previous space-time factorized attention (Bertasius et al., 2021). We also use relative positional encodings that are widely used to handle longer context length.

With this formulation, memory-augmented attentions are only computed together with each of $l/p_l$ patches that have the same spatial location (*i.e.*, temporal attention in video transformers (Bertasius et al., 2021)). This increase is not significant because the computation of attention is restricted only to the sample spatial locations. Our memory augmented attentions have $O(L^2 HW)$ computational complexity whereas full attention would scale as $O((LHW)^2)$.

Finally, recall that we build our architecture on W.A.L.T, but our general approach of using memory-augmented latent transformers can be applied more broadly to any video diffusion transformer architectures, such as Lu et al. (2023) and Ma et al. (2024b). We provide a detailed illustration of the architecture combined with W.A.L.T in Appendix B.

## 3 Related Work

We provide a brief discussion with important relevant literature. We provide more detailed discussion in Appendix C.

**Video diffusion models.** Inspired by the remarkable success of image diffusion models (Rombach et al., 2022; Chen et al., 2024b), many recent approaches have attempted to solve the challenging problem of video generation through diffusion models by extending ideas and architectures from the image domain (Blattmann et al., 2023; Gupta et al., 2023; Ho et al., 2022b; Harvey et al., 2022; He et al., 2022; Ge et al., 2023; Lu et al., 2023; Ma et al., 2024b; Ho et al., 2022a; Singer et al., 2023; Voleti et al., 2022; Weng et al., 2023; Yin et al., 2023; Yu et al., 2023b; 2024c; Zhou et al., 2022). Since memory and computation requirements increase dramatically due to the cubic complexity of videos as RGB pixels, most works have focused on efficient methods to generate videos via diffusion models. One avenue to achieve efficiency has been to train compact latent representations specialized for videos and training latent diffusion models in such spaces (Gupta et al., 2023; He et al., 2022; Yu et al., 2023b; 2024c). Other works have designed efficient model architectures to reduce computation in handling video data (Gupta et al., 2023; Bar-Tal et al., 2024; Yu et al., 2024c). Remarkably, with these efforts, very recent works have shown diffusion models can generate high-resolution (1080p) and quite long (up to 1 minute) videos if they use massive number of videos as data with an extremely large model (DeepMind, 2024; Brooks et al., 2024). Our method builds on both strands of prior work, leading to an efficient architecture specialized for and trained on *long* videos.

**Long video generation.** Many works have attempted to solve long video generation in different directions. First, there exist several works that interpret videos as compactly parameterized continuous functions of time (neural fields; (Sitzmann et al., 2020)) and train GANs (Goodfellow et al., 2014) to generate such function parameters (Yu et al., 2022b; Skorokhodov et al., 2022). These works have shown potential to generate arbitrarily long and smooth videos but are difficult to scale up with complex datasets due to the mode collapse problem of GANs (Srivastava et al., 2017). Another line of work has proposed an autoregressive approach to generate frames conditioned on previous context (Blattmann et al., 2023; Gupta et al., 2023; Yan et al., 2023; Weng et al., 2023; Ge et al., 2022; Villegas et al., 2023; Yan et al., 2021; Kondratyuk et al., 2023). However, due to the high-dimensionality of videos, these approaches condition on a small temporal window of previous frames. They also often suffer from error propagation leading to low-quality frames for longer temporal horizons (Huang et al., 2023), particularly on complex datasets. Several recent works have introduced training (or training-free) method to mitigate this issue (Ruhe et al., 2024; Kim et al., 2024a; Chen et al., 2024a; Xie et al., 2024; Lu et al., 2024b), but they still do not have long conteuxtual understanding capability. Other methods have proposed hierarchical generation to progressively generate long videos by interpolating between previously synthesized frames (Yin et al., 2023; Huang et al., 2023). However, if the target video length is extremely long, they need to generate very sparse video frames with very little local correlation, which is challenging, and they are also limited to generating frames up to the training length. MALT can be categorized as a diffusion based autoregressive approach; however, unlike the above approaches that condition on previously generated frames, we condition on memory, which is capable of capturing information going significantly further back in time, and is robust to error propagation when generating extremely long videos.

**Transformers for handling long context.** Recent works primarily from the large language models (LLMs) literature have proposed different ideas to handle extremely long contexts using transformers (Vaswani et al., 2017). First, several works have introduced efficient attention mechanisms (Wang et al., 2020; Kitaev et al., 2020; Beltagy et al., 2020; Choromanski et al., 2020) that reduce their quadratic complexity and thereby can handle longer contexts better. Next, some approaches have attempted to adopt recurrence in transformers (Dai et al., 2019; Bessonov et al., 2023; Bulatov et al., 2022; 2024; Hutchins et al., 2022; Peng et al., 2023), showing great potential to understand long contexts, sometimes even demonstrating their effectiveness on extremely long sequences (1M) (Bulatov et al., 2024). Some works introduce a method to cache previous keys and values from previous contexts (known as *KV-caches*) (Wu et al., 2022; Adnan et al., 2024; Pope et al., 2023), which are often combined with a relative positional encoding generalizable to a longer sequence than the training length (Su et al., 2024). Finally, there exist some approaches that implement efficient attention using hardware optimizations (Dao et al., 2022; Liu et al., 2023). Our method fits into this larger family of approaches by incorporating the recurrence technique into recent transformer-based diffusion approaches (Gupta et al., 2023; Peebles & Xie, 2023) to generate long sequences.

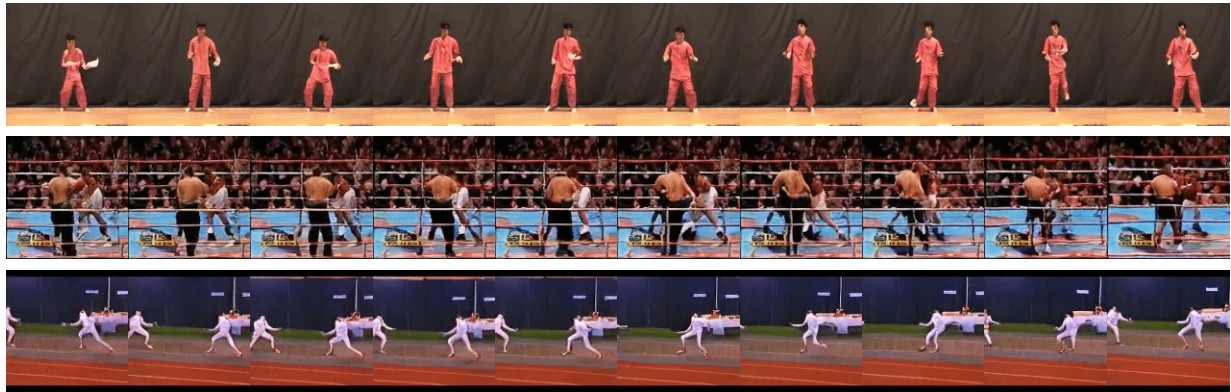

Figure 3: **Long video prediction results** on UCF-101. Each frame has 128×128 resolution. We visualize the frames with a stride of 16 (every 16 frames).

## 4 Experiments

We validate the performance of MALT and the effect of the proposed components through extensive experiments. In particular, we investigate the following questions:

- Can MALT generate or predict videos with a long horizon with understanding? (Table 1a, 1b, Figure 3, 4)
- Do the proposed components (recurrent memory, training for long-term stability) contribute to the final performance of images and videos (Table 2)?
- Is the frame quality of generated video maintained across frame indices? (Figure 5)

**Note.** Although we achieve strong performance on popular benchmarks, the focus of our paper is more on long-term stability rather than per-frame fidelity. Thus, the size of our models is relatively compact, and we do not train separate super-resolution modules to increase frame quality.

### 4.1 Setup

We explain some important setups in this section. We include more details in Appendix D and E.

**Datasets.** We use long video generation or prediction benchmarks (Skorokhodov et al., 2022; Yan et al., 2023) to evaluate our method. Specifically, we use UCF-101 (Soomro et al., 2012) for unconditional video *generation* with a length of 128 frames and use Kinetics-600 (Kay et al., 2017) for long video *prediction* to predict 80 frames conditioned on 20 frames. We choose these video datasets as they contain diverse and complex motions and thus can demonstrate the scalability of MALT to generalize to real-world complex videos. For text-to-video generation, we train on 89M text-short-video pairs (up to 37 frames) and 970M text-image pairs from public internet and internal sources. All datasets are center-cropped and resized to 128×128 resolution.

**Training details.** All models are trained with the AdamW (Loshchilov & Hutter, 2019) optimizer with a learning rate of 5$e$-5. For our autoencoder, we use the same architecture and configuration as W.A.L.T (Gupta et al., 2023). We also use similar diffusion model configurations that W.A.L.T used: specifically, we use the W.A.L.T XL-config for UCF-101 and W.A.L.T L-config for Kinetics-600 experiments. We also use the W.A.L.T L-config for text-to-video experiments. For the memory noise scale $\sigma_{\text{mem}}$, we use 0.1 in all experiments.

**Metrics.** We mainly use the Frèchet Video Distance (FVD (Unterthiner et al., 2018); lower is better) to evaluate the quality of generated videos, following recent works (Yu et al., 2023b; Skorokhodov et al., 2022; Kim et al., 2024b). On Kinetics-600, we also measure Peak signal-to-noise ratio (PSNR; higher is better), SSIM

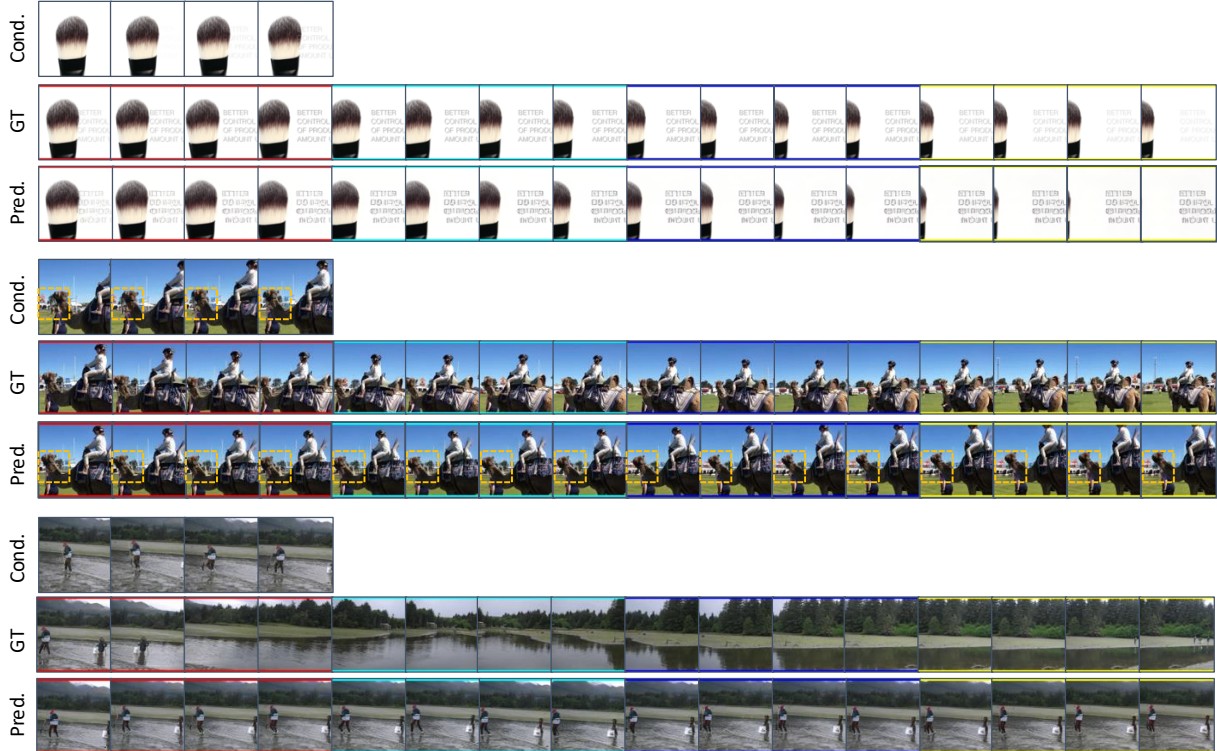

Figure 4: **Long video prediction results** on Kinetics-600. Each frame has 128×128 resolution. We visualize the frames with a stride of 5 (every 5 frames). The first 4 frames indicate an input condition. We mark the predicted frames, where the different color denotes different segments predicted from the model. For each video, we visualize the ground-truth in the first row and the prediction in the second row.

(higher is better), and perceptual metric (LPIPS (Zhang et al., 2018); lower is better) between ground-truth video frames and predicted frames to measure how well the prediction resembles the ground-truth.

**Baselines.** We use existing recent video generation and prediction methods as baselines that are capable of long video generation. Specifically, we use MoCoGAN (Tulyakov et al., 2018), MoCoGAN-HD (Tian et al., 2021), DIGAN (Yu et al., 2022b), StyleGAN-V (Skorokhodov et al., 2022), PVDM (Yu et al., 2023b), HVDM (Kim et al., 2024b), Latte (Ma et al., 2024b) with FreeNoise (Qiu et al., 2024), MeBT (Yoo et al., 2023), and CoordTok (Jang et al., 2024) with SIT-L/2 (Ma et al., 2024a) as baselines for long video generation. For the video prediction task, we consider the following baselines: Perceiver AR (Hawthorne et al., 2022), Latent FDM (Harvey et al., 2022), and TECO (Yan et al., 2023), which are recent generation methods designed for handling long videos with long-term understanding. We also compare with FIFO-diffusion (Kim et al., 2024a), (Henschel et al., 2025), and FreeNoise (Qiu et al., 2024), which are recent long video generation methods that are applied fto text-to-video situation.

## 4.2 Long-term contextual understanding

First, Figure 3 on UCF-101 shows that our method does understand long-term dependencies through memory latent vectors: in the video of two people fencing (last row), the first person on the left disappears entirely but is able to reappear in the last generated frames. Moreover, the videos in the first and second rows depict different actions and motions across the frames, rather than just static or repetitive patterns. This is also demonstrated in Figure 4 where we train on Kinetics-600. For instance, in the second video, there is an orange roof on the left side of the given prefix frames. The roof is covered by the camel's head for a few dozen frames before reappearing in the last predictions. Without a long understanding of the context, the model cannot remember these details, and the final predicted segment cannot contain this roof.

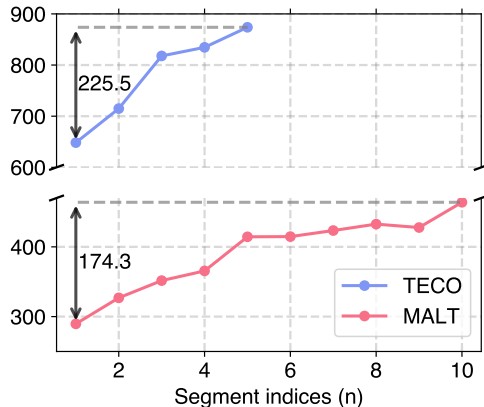

Figure 5: **Error propagation analysis.** FVD values on Kinetics-600 measured with a part of segments in the entire videos.

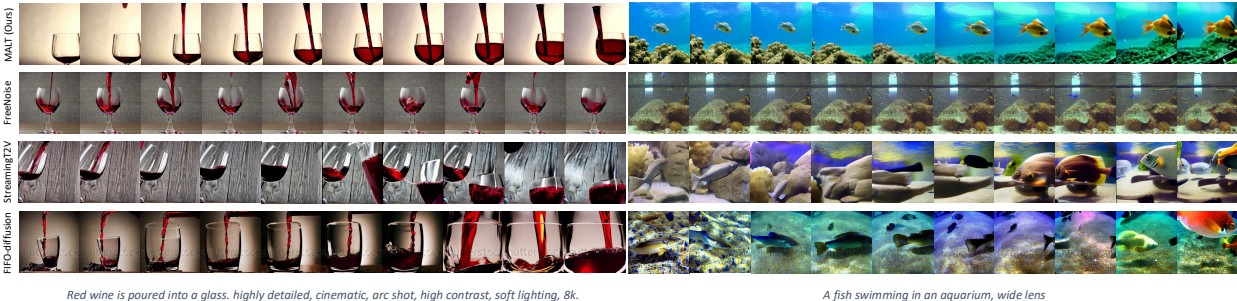

Figure 6: **Long text-to-video generation results**. We show two generated videos denoting frame indices within each frame. Each video is generated at 8 fps spanning $> 30$s and each frame has 128×128 resolution. Text prompts are provided under each visualization.

Note that this is possible due to the fundamental difference between predominant autoregressive methods and MALT: the former is "explicitly" autoregressive, while MALT is not. Specifically, prior works generate video frames conditioned on the previous explicit context (*e.g.*,, explicit in the sense of being conditioned on the last 8-16 RGB frames), whereas our method autoregressively generates video frames with respect to a "latent memory", which is capable of capturing much longer context in an efficient manner. This allows our model to have an extremely long receptive field without blowing up memory requirements.

### 4.3 Long-term stability

We also quantitatively analyze long term generation stability by visualizing FVD plots of the recent long-video generation training scheme, TECO, and MALT in Figure 5. Here, FVD values are computed by dividing the predicted video frames into multiple segments of 16-frame video clips and separately measuring FVD values using video clips at each specific interval. As shown in this figure, MALT generates higher-quality videos (*i.e.*, FVD values are always better at any given interval) but also suffers less from error accumulation issues than the previous state-of-the-art TECO. Specifically, the FVD difference between the first and the last FVD value of MALT is smaller (174.3) than TECO (225.5) even if the FVD difference of MALT is measured with 2× more number of segments (*i.e.*, MALT generates 2× more video frames). This indicates the frame quality of MALT drops slower than TECO, highlighting the robustness of MALT to error accumulation.

We also compare the text-to-video (T2V) generation results from recent training-free long T2V methods and MALT.[2] To compare this, we train a relatively small model (545M) compared to existing T2V models that usually have more than 2B parameters (Blattmann et al., 2023; Gupta et al., 2023; Ho et al., 2022a). As shown in Figure 6, MALT shows reasonable T2V generation on complex text prompts. More importantly, generated frames are consistent and stable over a long temporal horizon, while other baselines fail to achieve this, *e.g.*, either the wine is not consistently poured, or the wine glasses are not consistently maintained.. We provide more visualizations in Appendix G: we show that MALT can be generalized to generate up to 2 minute videos with 8 fps.

Table 1: **System-level comparision.** (a) FVDs of video generation and models on UCF-101 and (b) FVD, PSNR, SSIM and LPIPS of video prediction models on Kinetics-600. For FVD on Kinetics-600, we measure it with entire video frames, namely including initial ground-truth frames given as contexts and 80 predicted frames from each baseline. For the other three metrics, we measure the values using only the predicted 80 frames. Bold indicate the best results.

(a) UCF-101

| Method | FVD $\downarrow$ |
|---|---|
| MoCoGAN (Tulyakov et al., 2018) | 3679.0 |
| + StyleGAN2 (Karras et al., 2020) | 2311.3 |
| MoCoGAN-HD (Tian et al., 2021) | 2606.5 |
| DIGAN (Yu et al., 2022b) | 2293.7 |
| StyleGAN-V (Skorokhodov et al., 2022) | 1773.4 |
| Latte (Ma et al., 2024b) + FreeNoise (Qiu et al., 2024) | 1157.7 |
| MeBT (Yoo et al., 2023) | 968 |
| PVDM (Yu et al., 2023b) | 505.0 |
| HVDM (Kim et al., 2024b) | 549.7 |
| CoordTok (Jang et al., 2024) + SiT-L/2 (Ma et al., 2024a) | 369.3 |
| **MALT (Ours)** | **220.4** |

(b) Kinetics-600

| Method | FVD $\downarrow$ | PSNR $\uparrow$ | SSIM $\uparrow$ | LPIPS $\downarrow$ |
|---|---|---|---|---|
| Perceiver AR (Hawthorne et al., 2022) | 1022 | 13.4 | 0.310 | 0.404 |
| Latent FDM (Harvey et al., 2022) | 960 | 13.2 | 0.334 | 0.413 |
| TECO (Yan et al., 2023) | 799 | 13.8 | 0.341 | 0.381 |
| **MALT (Ours)** | **392** | **15.4** | **0.437** | **0.276** |

## 4.4 Quantitative results

Table 1a and 1b summarize long video generation and prediction results on UCF-101 and Kinetics-600 datasets, respectively. On UCF-101, MALT achieves an FVD score (lower is better) of 220.4, significantly outperforming existing video generation methods including unconditional latent video diffusion models capable of long video generation, such as PVDM (Yu et al., 2023b) and HVDM (Kim et al., 2024b). Similarly, on a long video prediction task on Kinetics-600, MALT shows much better performance than the previous state-of-the-art method, TECO, on all four metrics despite have $\sim 2\times$ fewer parameters (440M for MALT and 1.1B for TECO).

## 4.5 Ablation studies

We conduct ablation studies and report in Table 2. We also perform error propagation analysis and summarize the result in Figure 5. For further analysis, see Appendix F.

---

[2]There are numerous factors that affect the frame quality, to name a few, model size, text caption fidelity, and video-text alignment. Thus, an apples-to-apples comparison of videos based on frame fidelity is difficult because each method uses design choices for aforementioned factors: Thus, we focus on comparing long-term stability of generated video frames.

Table 2: **Ablation studies.** FVD, PSNR, and SSIM values on Kinetics-600. Gray indicates that the component is fixed for ablation. "Last only" a condition model only on the very previous segment. "kv cache": a conditional model that predicts a segment on all previous segments without any compression. "Recurrent": our recurrent memory scheme that uses fixed-size hidden states as condition. "Robust": our training technique for long-term stability. W.A.L.T-L indicates the baseline MALT model with all MALT features removed.

|  |  | Robust | FVD | PSNR | SSIM |
|---|---|---|---|---|---|
| W.A.L.T-L (Gupta et al., 2023) |  |  | 459 | 14.8 | 0.400 |
| Memory design | Last only | ✓ | 423 | 15.0 | 0.424 |
|  | kv-cache | ✓ | 396 | 15.1 | 0.429 |
|  | Recurrent (ours) | ✓ | 392 | 15.4 | 0.437 |
| Training technique | Recurrent |  | **383** | 14.7 | 0.401 |
|  | Recurrent (ours) | ✓ | 392 | **15.4** | **0.437** |

**Memory design.** As shown in Table 2, a conditional model that only uses the previous segment ("Last only") shows worse performance than other models that use longer contexts, indicating the importance of long-term contextual understanding. Moreover, our recurrent and compact memory design ("Recurrent") shows comparable performance compared to the kv-cache that uses the entire context without any compression, which speaks to the effectiveness of MALT.

**Training objective.** The model trained without noise augmentation and correlated prior distribution shows worse SSIM and PSNR (*e.g.*, PSNR gets worse from 15.4 to 14.7), exhibiting the frame quality drop across the frame indices. It demonstrates the importance of our training techniques that use noise augmentation to the memory latent vector and correlated prior distribution for stabilizing the frame quality.

## 5 Conclusion

We proposed MALT Diffusion, a latent video diffusion model for any-length video generation. It is based on proposing a new DiT architecture, which employs a memory for encoding long video context into succinct low-dimensional latent vectors. Our approach is general and can be applied in principle to any architectures and we believe it will facilitate long video synthesis breaking beyond the current limits. We discuss social impact and limitations in Appendix H.

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

## A    Training and Sampling Procedure

We provide detailed training and sampling procedure of MALT in Algorithm 1 and 2, respectively.

---

**Algorithm 1** MALT Diffusion (Training)

---

1: **while** Converge **do**
2:     Sample the number of segments $N$ in $[1, N_{\max}]$.
3:     $\mathbf{h}^0 \leftarrow [\mathbf{0}, \ldots, \mathbf{0}]$
4:     **for** $n = 0$ to $N - 2$ **do**          ▷ *Compute memory latent vector that encodes from segment 1 to $N-1$.*
5:         $\mathbf{z}^{n+1} \leftarrow F(\mathbf{x}^{n+1})$.
6:         **if** $n < N - 2$ **then**
7:             $\mathbf{h}^{n+1} \leftarrow \text{HiddenState}\big(D_{\boldsymbol{\theta}}(\mathbf{z}^{n+1}, 0; \text{sg}(\mathbf{h}^n), \mathbf{c})\big)$
8:         **else**
9:             $\boldsymbol{\xi} \sim \mathcal{N}(\mathbf{0}, \sigma_{\text{mem}}^2 \mathbf{I})$
10:             $\tilde{\mathbf{h}}^{n+1} \leftarrow \text{HiddenState}\big(D_{\boldsymbol{\theta}}(\mathbf{z}^{n+1} + \boldsymbol{\xi}, 0; \text{sg}(\mathbf{h}^n), \mathbf{c})\big)$
11:         **end if**
12:     **end for**
13:     $\mathbf{z}^N \leftarrow F(\mathbf{x}^N)$, $\boldsymbol{\epsilon} \sim \mathcal{N}(\mathbf{0}, \mathbf{I})$, $t \sim [0, T]$.          ▷ *Loss computation.*
14:     Compute $\mathbf{z}_t^N$ using the forward SDE with $\mathbf{z}^N$ and $\boldsymbol{\epsilon}$.
15:     Compute loss $\mathcal{L}(\boldsymbol{\theta}) := \mathbb{E}\Big[\lambda(t) \|D_{\boldsymbol{\theta}}(\mathbf{z}_t^N, t; \tilde{\mathbf{h}}^{N-1}, \mathbf{c}) - \mathbf{z}_0^N\|_2^2\Big]$,
16:     Update $\boldsymbol{\theta}$ using a gradient descent.
17: **end while**

---

**Algorithm 2** MALT Diffusion (Sampling)

---

1: **for** $n = 1$ to $N$ **do**          ▷ *Autoregressively generate $n$-th latent vector $\mathbf{z}^n$.*
2:     Sample the random noise $\mathbf{z}_T^n \sim p(\mathbf{z}_T)$.
3:     **for** $i$ in $\{0, \ldots, M-1\}$ **do**
4:         Compute the score $\boldsymbol{\epsilon}_i \leftarrow D_{\boldsymbol{\theta}}(\mathbf{z}_{t_i}^n, t; \mathbf{h}^{n-1}, \mathbf{c})$.
5:         Compute $\mathbf{z}_{t_{i+1}}^n \leftarrow \mathbf{z}_{t_i}^n + (t_{i+1} - t_i)\boldsymbol{\epsilon}_i$          ▷ *Euler solver; can be different with other solvers.*
6:     **end for**
7:     $\mathbf{h}^n = \text{HiddenState}\big(D_{\boldsymbol{\theta}}(\mathbf{z}_M^n, 0; \mathbf{h}^{i-1}, \mathbf{c})\big)$          ▷ *Compute memory latent vector.*
8: **end for**
9: Decode $[\mathbf{x}^1, \ldots, \mathbf{x}^N]$ from generated latent vectors $[\mathbf{z}^1, \ldots, \mathbf{z}^N]$.
10: Output the generated video $[\mathbf{x}^1, \ldots, \mathbf{x}^N]$.

---

## B    Architecture Illustration

Figure 7 visualizes the MALT architecture in detail as applied to the existing W.A.L.T (Gupta et al., 2023) architecture. W.A.L.T is a variant of diffusion transformer (DiT) (Peebles & Xie, 2023) that uses repeating spatiotemporal and spatial window attention layers instead of full attention. As illustrated in this figure, MALT adds memory attention layers at the beginning of each spatiotemporal window attention layer and spatial window attention layer. Here, window attention layers are similar to the original DiT that uses adaptive instance normalization (AdaIN), where we use LoRA Hu et al. (2021) for parametrization of MLP proposed by W.A.L.T to reduce the number of parameters without performance degradation. For memory attention layer, we simply add a cross attention layer without AdaIN.

## C   More Discussion with Related Work

**Diffusion model for sequential data.** We emphasize that although we primarily explore video data in this paper, our method is not limited to video data. There exists multiple works that try to generate general sequential data (*e.g.*, climate data, PDE data, audio, *etc.*) through diffusion models. For instance, Ruhe et al. (2024) formulates a diffusion process specialized for sequential data; they propose denoising process with a *sliding window*, where the noise scale of elements within a window is set differently by considering uncertainty of each element. Another line of work (Ge et al., 2023; Lu et al., 2024a) have tried to solve this problem by exploring temporal correlation of Gaussian noise in diffusion process with *fixed-size* sequences. Lippe et al. (2023) tries to solve PDE through diffusion model based on autoregressive rollout, and demonstrates high-frequency details can be predicted better than existing PDE solvers. Our method primarily focuses on generating video data, where each element (*i.e.*, video frame) is already quite high-dimensional and thus mitigating error propagation and ensuring large context window size is more difficult than other temporal data. Therefore, we believe our method can be extending to other forms of sequential data listed above.

**Video generation.** There are many works to tackle the problem of video generation. First, there are approaches that uses generative adversarial network (GAN; Goodfellow et al. (2014)) for modeling video distribution (Tulyakov et al., 2018; Yu et al., 2022b; Skorokhodov et al., 2022; Tian et al., 2021; Acharya et al., 2018; Clark et al., 2019; Fox et al., 2021; Gordon & Parde, 2021; Kahembwe & Ramamoorthy, 2020; Munoz et al., 2021; Saito et al., 2017; Vondrick et al., 2016; Yushchenko et al., 2019). They usually extend popular image GAN architectures (*e.g.*, StyleGAN (Karras et al., 2020)) by considering an additional temporal axis. Another line of works encode videos in discrete token space using popular vector-quantized autoencoders (Yu et al., 2024a; van den Oord et al., 2017; Yu et al., 2022a) and learn video distribution in this latent space, either with autoregressive transformers (Yu et al., 2024a; Ge et al., 2022; Yan et al., 2021; Kalchbrenner et al., 2017; Rakhimov et al., 2020; Weissenborn et al., 2020) or masked generative transformers (Yoo et al., 2023; Yu et al., 2023a). Finally, many of recent works propose diffusion-based approach for generating videos (Ho et al., 2022b; Harvey et al., 2022; Lu et al., 2023; Singer et al., 2023; Weng et al., 2023; Höppe et al., 2022; Yang et al., 2022), and some of them have attempt to exploit knowledge learned from large-scale image datasets (Blattmann et al., 2023; He et al., 2022; Ge et al., 2023; Ho et al., 2022a; Singer et al., 2023; Wang et al., 2023; An et al., 2023) by fine-tuning image diffusion models or do image-video joint training. Our model is also categorized as diffusion-based approach to synthesize long videos.

**Video prediction.** As our model has a capability of both video generation and prediction, it also has a close relationship to existing video prediction methods (Srivastava et al., 2015; Finn et al., 2016; Denton & Birodkar, 2017; Babaeizadeh et al., 2018; Denton & Fergus, 2018; Lee et al., 2018; Villegas et al., 2019; Kumar et al., 2020; Franceschi et al., 2020; Luc et al., 2020; Lee et al., 2021; Seo et al., 2022). These methods usually use given frames as condition to the model and generate future frames via popular deep generative models, such as GANs, diffusion models, and autoregressive long video prediction (Harvey et al., 2022; Yan et al., 2023), in contrast to previous works that usually focus on predicting a few video frames. Our method also shares a similar goal to predict video frames with a long horizon.

## D   Setup

### D.1   Datasets

**UCF-101.** UCF-101 (Soomro et al., 2012) is a video dataset widely used for evaluation of video generation methods. Is consists of 101 classes of different human actions, where each video has $320 \times 240$ resolution frames. The dataset includes 13,320 videos with 9,537 training split and 3,783 test split. Following the previous recent video generation literature we only use train split and test split for evaluation (Yu et al., 2023a; Singer et al., 2023). We center-cropped each video and resized it into $128 \times 128$ resolution. We use first 128 frames of videos for training and evaluation.

**Kinetics-600.** Kinetics-600 (Kay et al., 2017) is a large-scale complex video dataset consisting of 600 action categories. It contains about 480,000 videos in total, where they are divided into 390,000, 30,000, and 60,000 videos for train, validation, and test splits (respectively). Following the previous video prediction

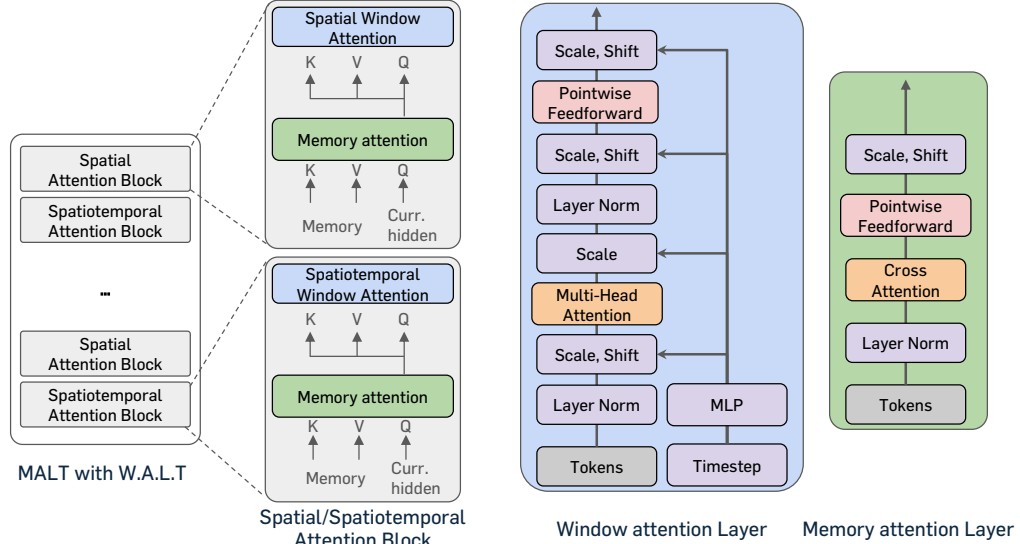

Figure 7: **Detailed architecture illustration** of MALT if it is incoporated with W.A.L.T (Gupta et al., 2023).

benchmarks (Gupta et al., 2023; Yan et al., 2023; Yu et al., 2023a), we center-crop each video frame and resize each video frame as 128×128 resolution. Due to the different tokenizer used in our method, we use 17 frames for condition (a bit different from TECO (Yan et al., 2023) that uses 20 frames for condition) and but predict 80 frames same as TECO. We use train split for train and validation split for evaluation.

**T2V dataset.** We use 89M text-short-video pairs (up to 37 frames) and 970M text-image pairs from public and internal sources. All datasets are center-cropped and resized to 128×128 resolution. For videos, we sample the frames with a fps of 8.

## D.2 Hyperparameters

We mostly follow the same hyperparameter setup to W.A.L.T (Gupta et al., 2023). Specifically, we provide the detailed hyperparameter values in Table 3.

## D.3 Metrics

For Fréchet video distance (FVD; Unterthiner et al. (2018)), we use the same protocol used in previous popular video generation works (Yu et al., 2022b; Tulyakov et al., 2018); we use a pretrained I3D network (Tran et al., 2015) to compute feature for evaluating statistics to compute FVD. For UCF-101, we use 2,048 real and fame samples following the common practice in previous video generation literature (Yu et al., 2022b; Skorokhodov et al., 2021; Yu et al., 2023b). For Kinetics-600, we use 256 samples for FVD evaluation and average the values of 4 runs, following the setup in TECO (Yan et al., 2023). For other metrics (PSNR, SSIM, LPIPS) used in evaluation on Kinetics-600, we also follow the evaluation setup of TECO: we compute them in per-frame manner between predicted frames and ground-truth frames, and then average them.

## D.4 Training resources

All experiments are conducted with Google Cloud TPU v5e 16×16 instances, where each chip has a 16GB HBM2 capacity. Note that training can be done on devices with relative small memory (less than 16GB) because of our efficient training scheme design.

Table 3: Hyperparameter setup.

| | UCF-101 | Kinetics-600 | T2V |
|---|---|---|---|
| **Autoencoder** | | | |
| Input dim. | 17×128×128×3 | 17×128×128×3 | 17×128×128×3 |
| $d_s$, $d_t$ | 8, 4 | 8, 4 | 8, 4 |
| Channel dim. | 128 | 128 | 128 |
| Channel multiplier | 1, 2, 2, 4 | 1, 2, 2, 4 | 1, 2, 2, 4 |
| Training duration | 2,000 epochs | 270,000 steps | 1,000,000 steps |
| Batch size | 256 | 256 | 256 |
| lr scheduler | Cosine | Cosine | Cosine |
| Optimizer | Adam | Adam | Adam |
| **Diffusion model** | | | |
| Input dim. | 5×16×16 | 5×16×16 | 5×16×16 |
| Num. layers | 28 | 24 | 24 |
| Hidden dim. | 1,152 | 1,024 | 1,024 |
| Num. heads | 16 | 16 | 16 |
| $\sigma_{\mathrm{mem}}$ | 0.1 | 0.1 | 0.1 |
| $N$ | 7 | 5 | 2 |
| Training duration | 120,000 steps | 270,000 steps | 700,000 steps |
| Batch size | 256 | 256 | 256 |
| lr scheduler | Cosine | Cosine | Cosine |
| Optimizer | AdamW | AdamW | AdamW |
| lr | 0.0005 | 0.0005 | 0.0002 |
| **Diffusion** | | | |
| Diffusion steps | 1000 | 1000 | 1000 |
| Noise schedule | Linear | Linear | Linear |
| $\beta_0$ | 0.0001 | 0.0001 | 0.0001 |
| $\beta_T$ | 0.02 | 0.02 | 0.02 |
| Training objective | v-prediction | v-prediction | v-prediction |
| Sampler | DDIM (Song et al., 2021a) | DDIM (Song et al., 2021a) | DDIM (Song et al., 2021a) |
| Sampling steps | 50 | 50 | 50 |
| Guidance | - | - | ✓ |

## E Baselines

In what follows, we explain the main idea of baseline methods that we used for the evaluation.

- **MoCoGAN** (Tulyakov et al., 2018) proposes a video GAN to generate videos by decomposing motion and content of videos into two different latent vectors.

- **MoCoGAN-HD** (Tian et al., 2021) also proposes a video GAN based on motion-content decomposition but uses a latent space of pretrained image GAN to achieve the goal.

- **DIGAN** (Yu et al., 2022b) proposes to represent videos as implicit neural representations (INRs) (Sitzmann et al., 2020) and introduces a GAN to generate such INR parameters.

- **StyleGAN-V** (Skorokhodov et al., 2022) interprets videos as continuous function of time $t$ and extend StyleGAN-2 (Karras et al., 2020) architecture to efficiently learn long video distribution.

- **PVDM** (Yu et al., 2023b) proposes a latent diffusion model based on triplane-based encoding of videos (Kim et al., 2022) to avoid usage of computational-heavy 3D convolutions.

- **HVDM** (Kim et al., 2024b) uses the ideas in PVDM but also incorporates 3D wavelet representation for video encoding to achieve better video reconstruction and generation.

- **(Latent) FDM** (Harvey et al., 2022) proposes a diffusion model framework for long videos, by exploring various schemes to choose frames to be noised in the target long video tensor.

- **Perceiver AR** (Hawthorne et al., 2022) proposes an autoregressive model that can handle long contexts in efficient and in domain-agnostic manner.

- **CoordTok** (Jang et al., 2024) presents a continuous video tokenizer that can encode long videos into compact triplane latent representations. They also show this latent representations greatly improve generation efficiency and efficacy.

- **TECO** (Yan et al., 2023) proposes a masked generative transformer (Chang et al., 2022) specialized for long video generation, based on additional compression of image latent vector from image VQGAN (Yu et al., 2022a) and causal transformer to encode these compressed sequences.

## F   Additional Analysis

Table 4: **Additional analysis.** FVD, PSNR, and SSIM values on Kinetics-600. Gray indicates that the component is fixed for ablation of other components.

|  |  |  | FVD | PSNR | SSIM |
|---|---|---|---|---|---|
| $\sigma_{\mathrm{mem}}$ | 0.2 | 97 | 382 | 15.0 | 0.426 |
|  | 0.1 | 97 | 392 | 15.4 | 0.437 |
| Training | 0.1 | 57 | 437 | 14.9 | 0.421 |
| Length | 0.1 | 97 | 392 | 15.4 | 0.437 |

In addition to experiments in Section 4.5, we perform several additional experiments to show the effect of hyperparameter $\sigma_{\mathrm{mem}}$ and the length of video (*i.e.*, number of segments) used in training, and report these results in Table 4. As shown in this table, MALT is quite robust the value choice of $\sigma_{\mathrm{mem}}$, and training MALT on longer videos improves the model to generate longer context.

## G   Additional Qualitative Results

We provide video generation results from MALT, mostly on long text-to-video generation results.

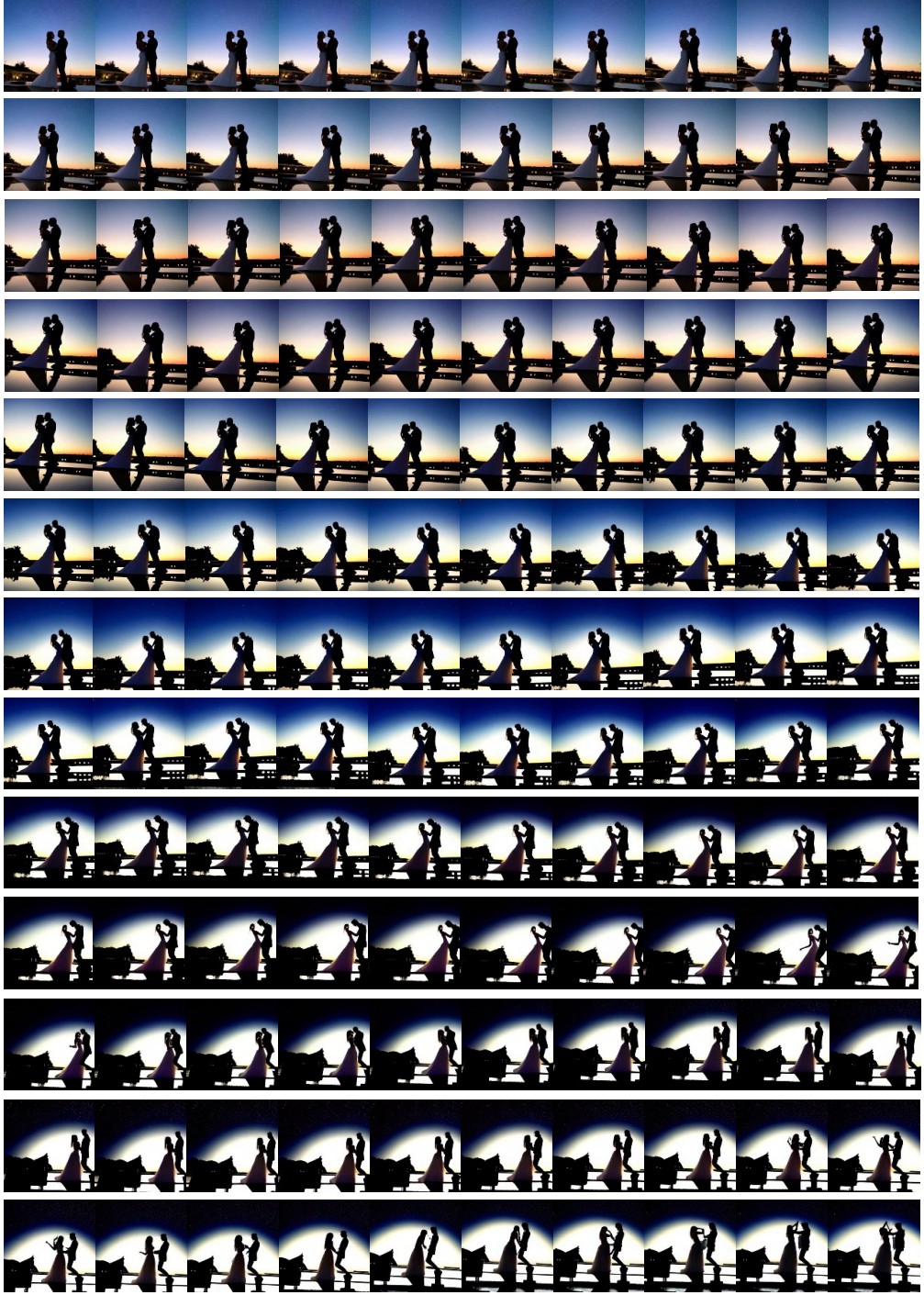

*A couple dancing under the stars on a rooftop*

Figure 8: **Long text-to-video generation results** from MALT. We visualize video frames with a stride of 5. We visualize first 650 frames here and the next 600 frames are visualized in Figure 9.

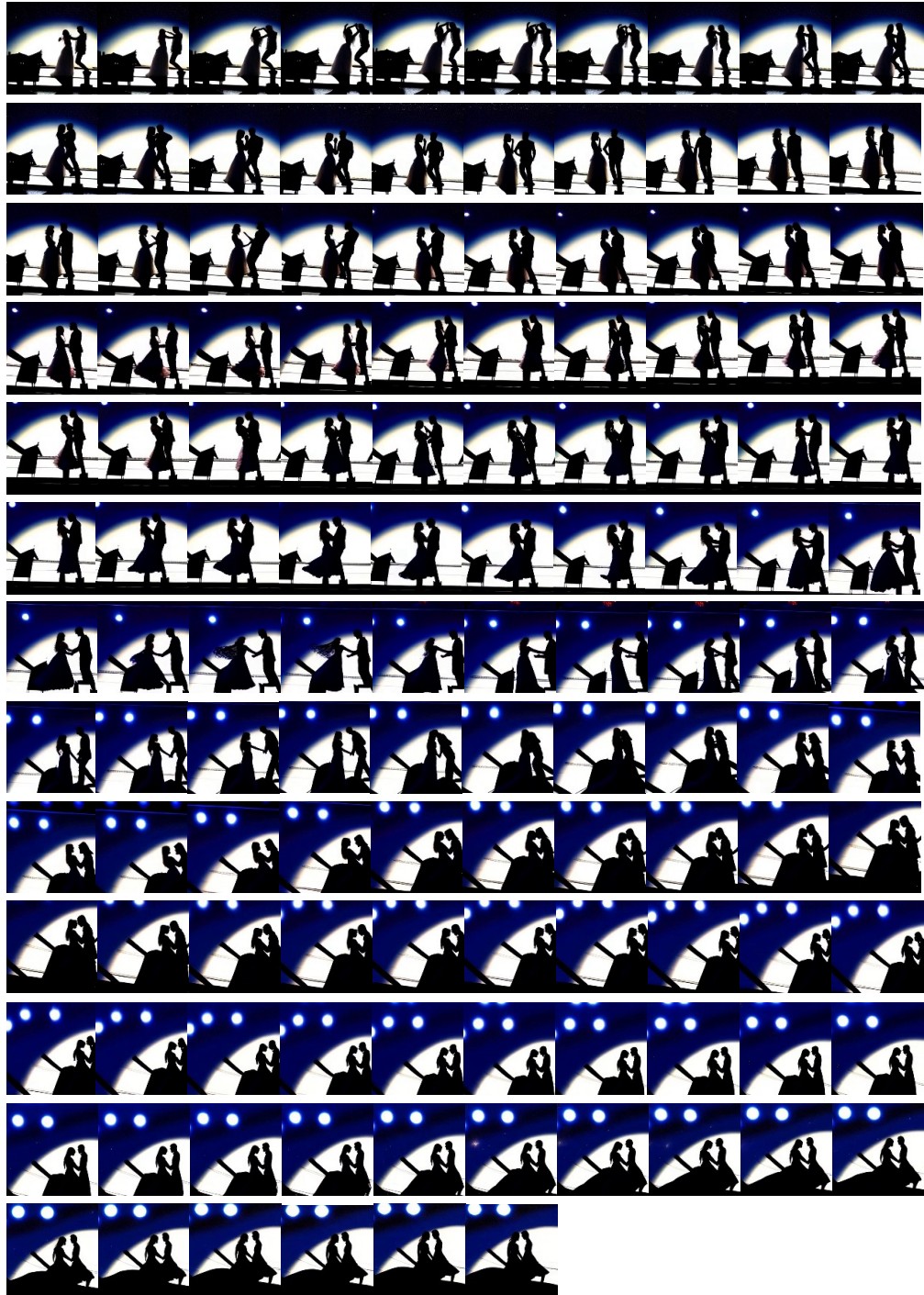

*A couple dancing under the stars on a rooftop*

Figure 9: **Long text-to-video generation results** from MALT. We visualize video frames with a stride of 5. Video frames are continued from Figure 8.

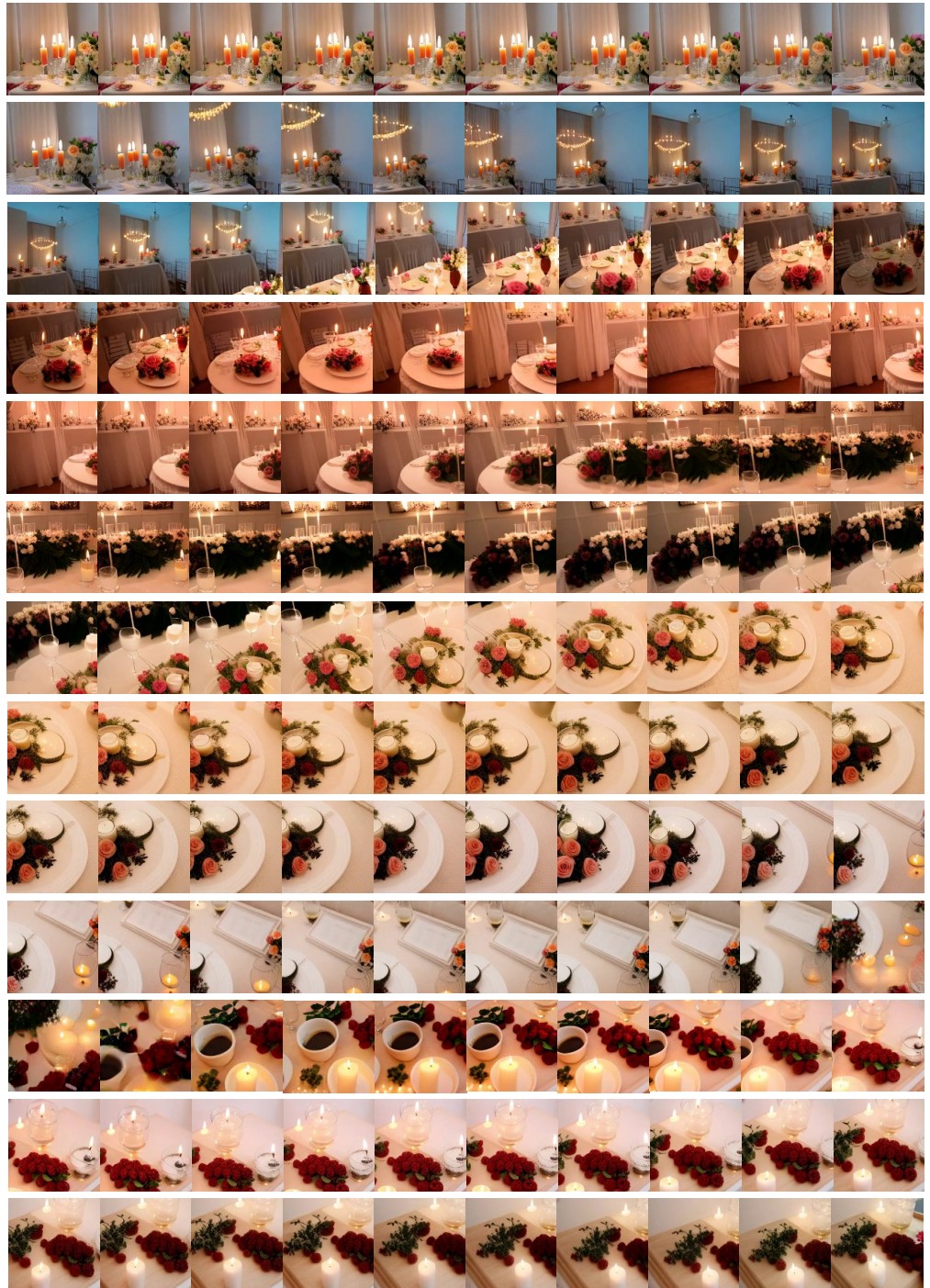

*A candlelit dinner with with table for two, flowers, candles, and wine*

Figure 10: **Long text-to-video generation results** from MALT. We visualize video frames with a stride of 5. We visualize first 650 frames here and the next 600 frames are visualized in Figure 11.

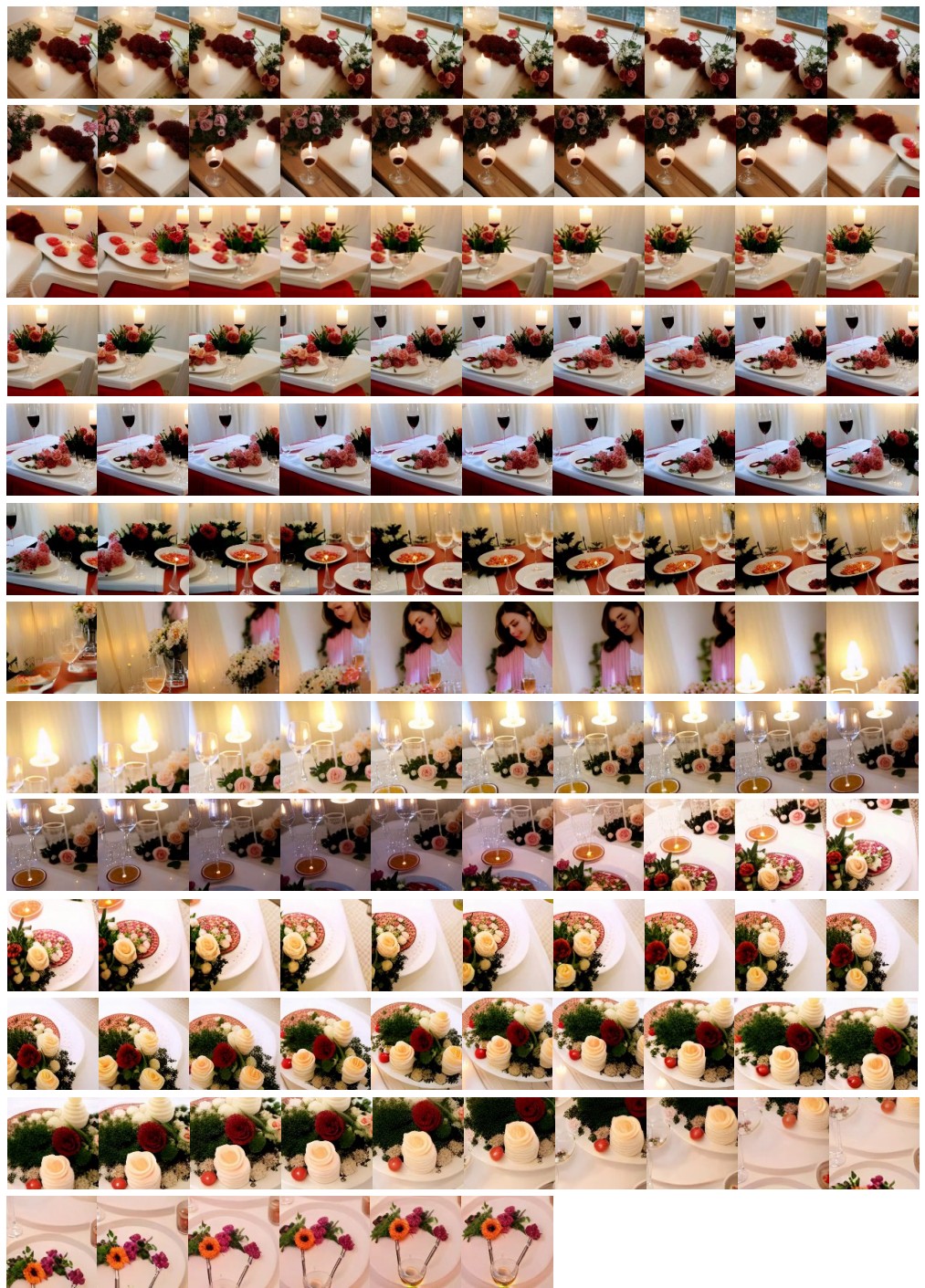

*A candlelit dinner with with table for two, flowers, candles, and wine*

Figure 11: **Long text-to-video generation results** from MALT. We visualize video frames with a stride of 5. Video frames are continued from Figure 10.

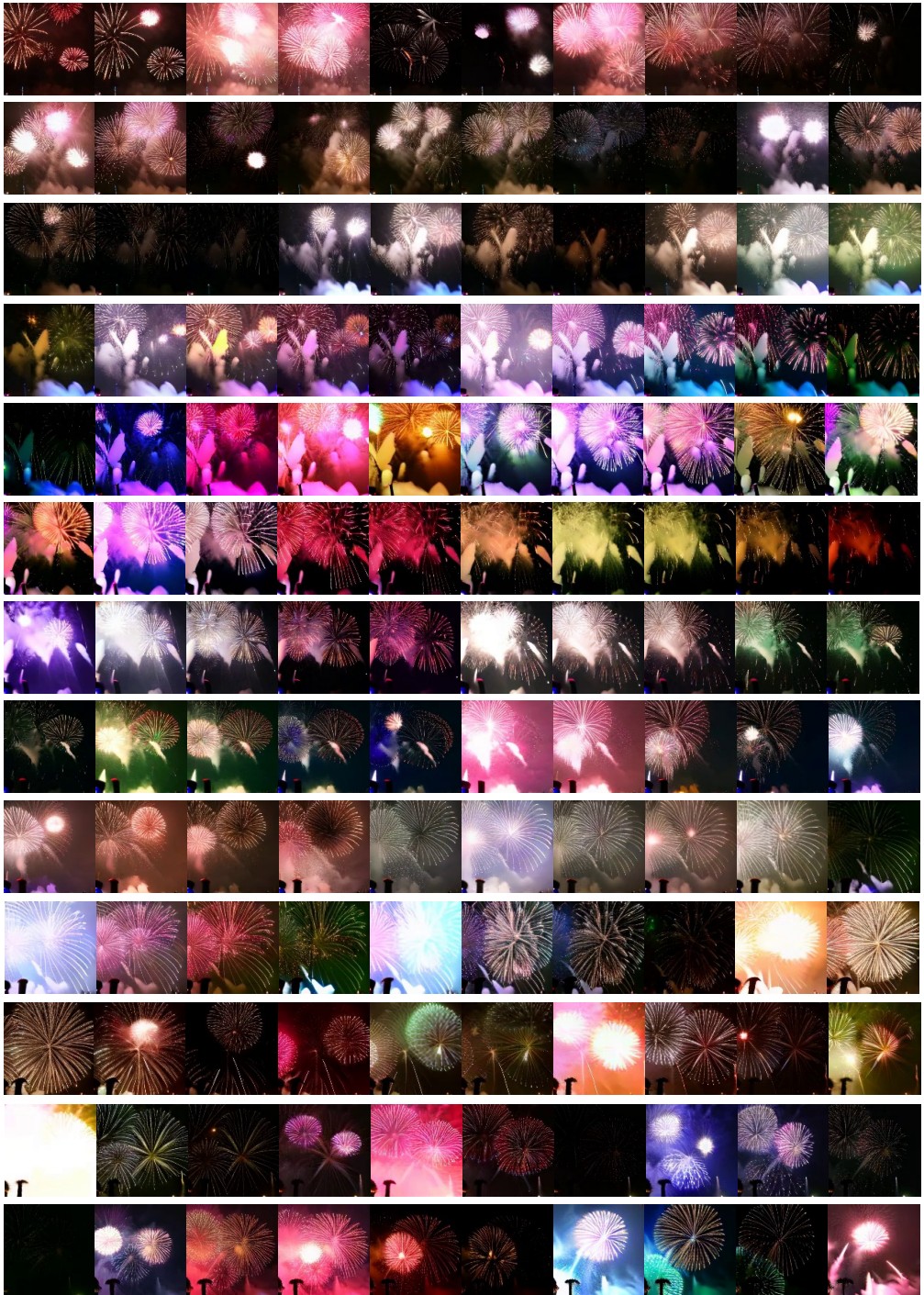

*Fireworks display at Disneyworld, highly detailed, cinematic, high contrast, soft lighting, 8k.*

Figure 12: **Long text-to-video generation results** from MALT. We visualize video frames with a stride of 5. We visualize first 650 frames here and the next 600 frames are visualized in Figure 13.

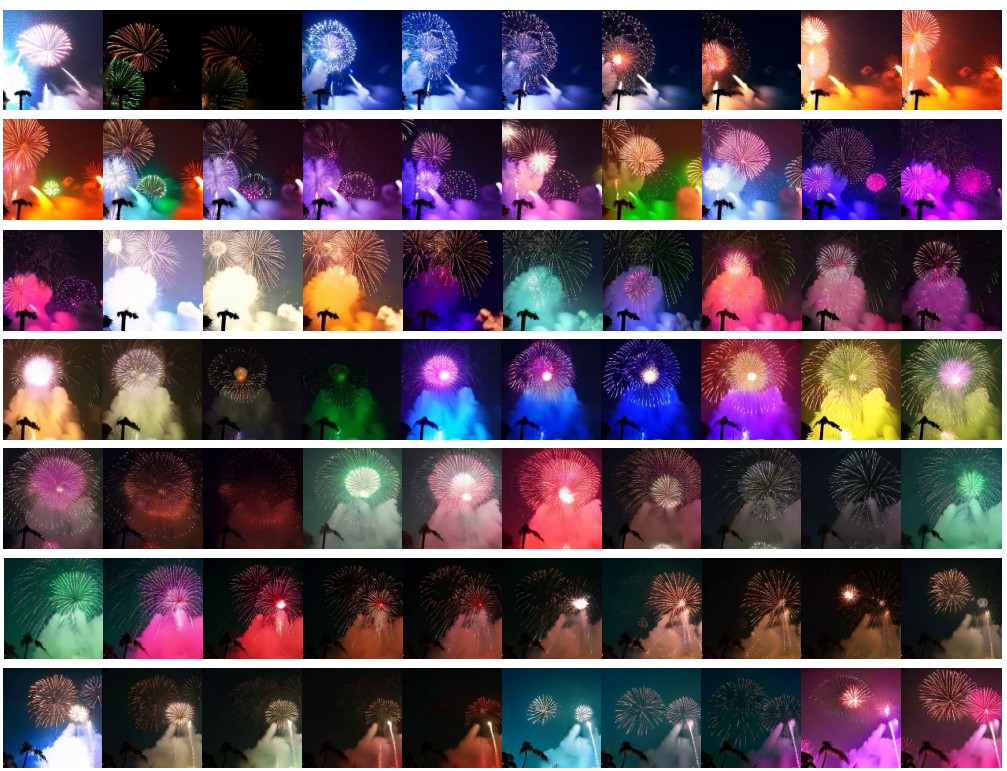

*Fireworks display at Disneyworld, highly detailed, cinematic, high contrast, soft lighting, 8k.*

Figure 13: **Long text-to-video generation results** from MALT. We visualize video frames with a stride of 5. Video frames are continued from Figure 12.

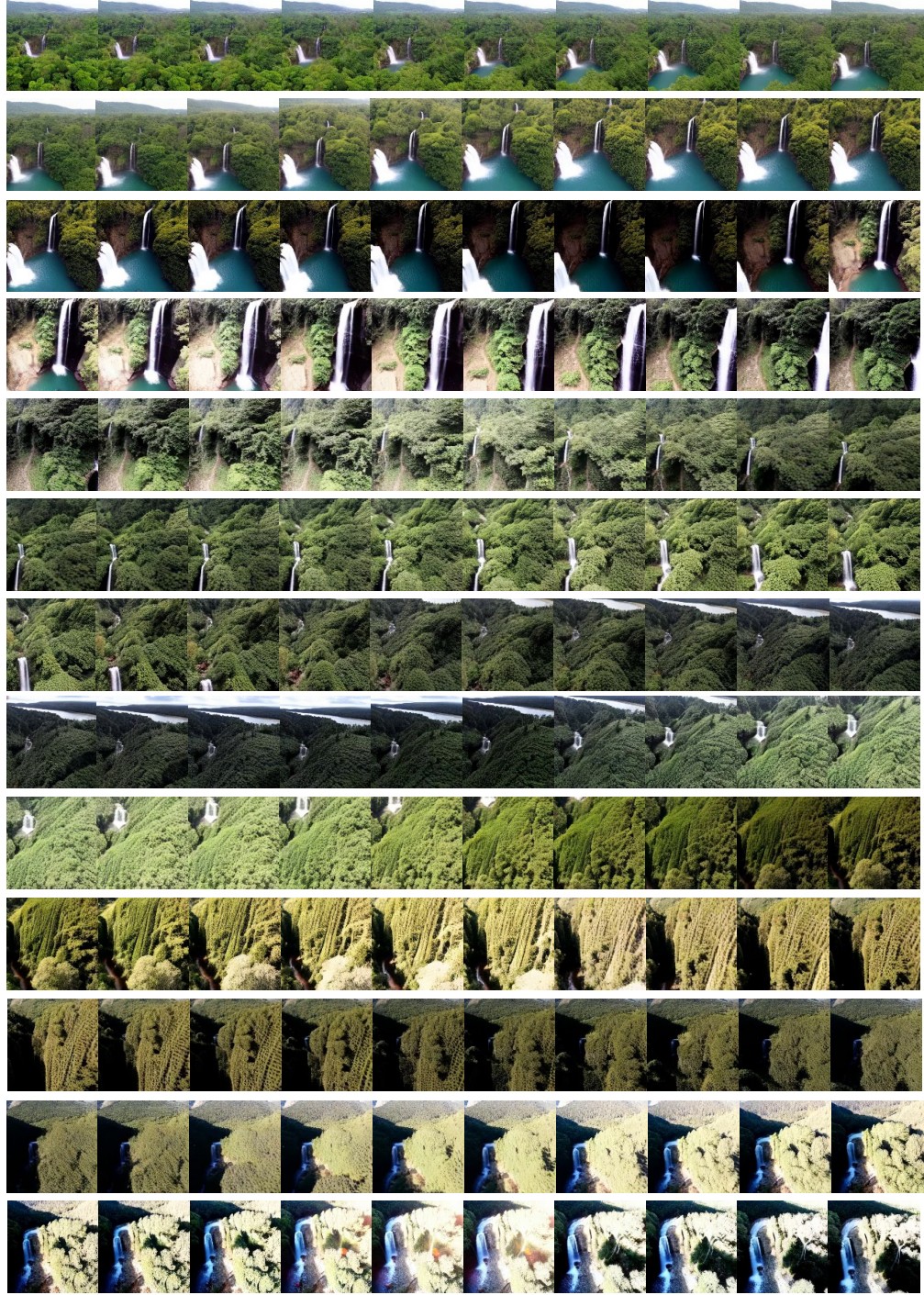

*A jungle with trees, vines, animals, and waterfall, aerial view.*

Figure 14: **Long text-to-video generation results** from MALT. We visualize video frames with a stride of 5. We visualize 650 frames in total.

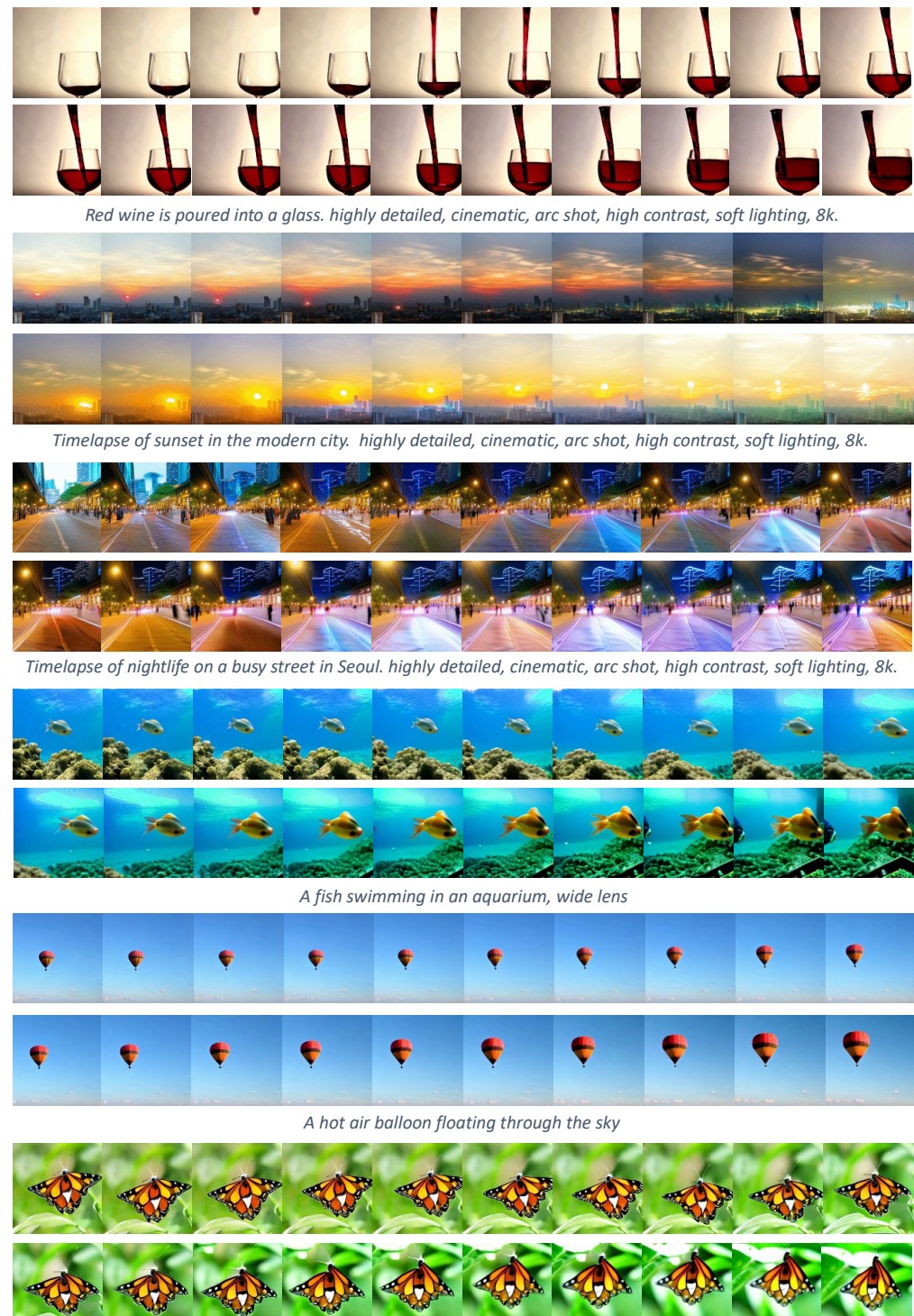

*Red wine is poured into a glass. highly detailed, cinematic, arc shot, high contrast, soft lighting, 8k.*

*Timelapse of sunset in the modern city. highly detailed, cinematic, arc shot, high contrast, soft lighting, 8k.*

*Timelapse of nightlife on a busy street in Seoul. highly detailed, cinematic, arc shot, high contrast, soft lighting, 8k.*

*A fish swimming in an aquarium, wide lens*

*A hot air balloon floating through the sky*

*Butterfly fluttering its wings, slow motion.*

Figure 15: **Text-to-video generation results** from MALT. We visualize video frames with a stride of 5. We visualize 100 frames in total.

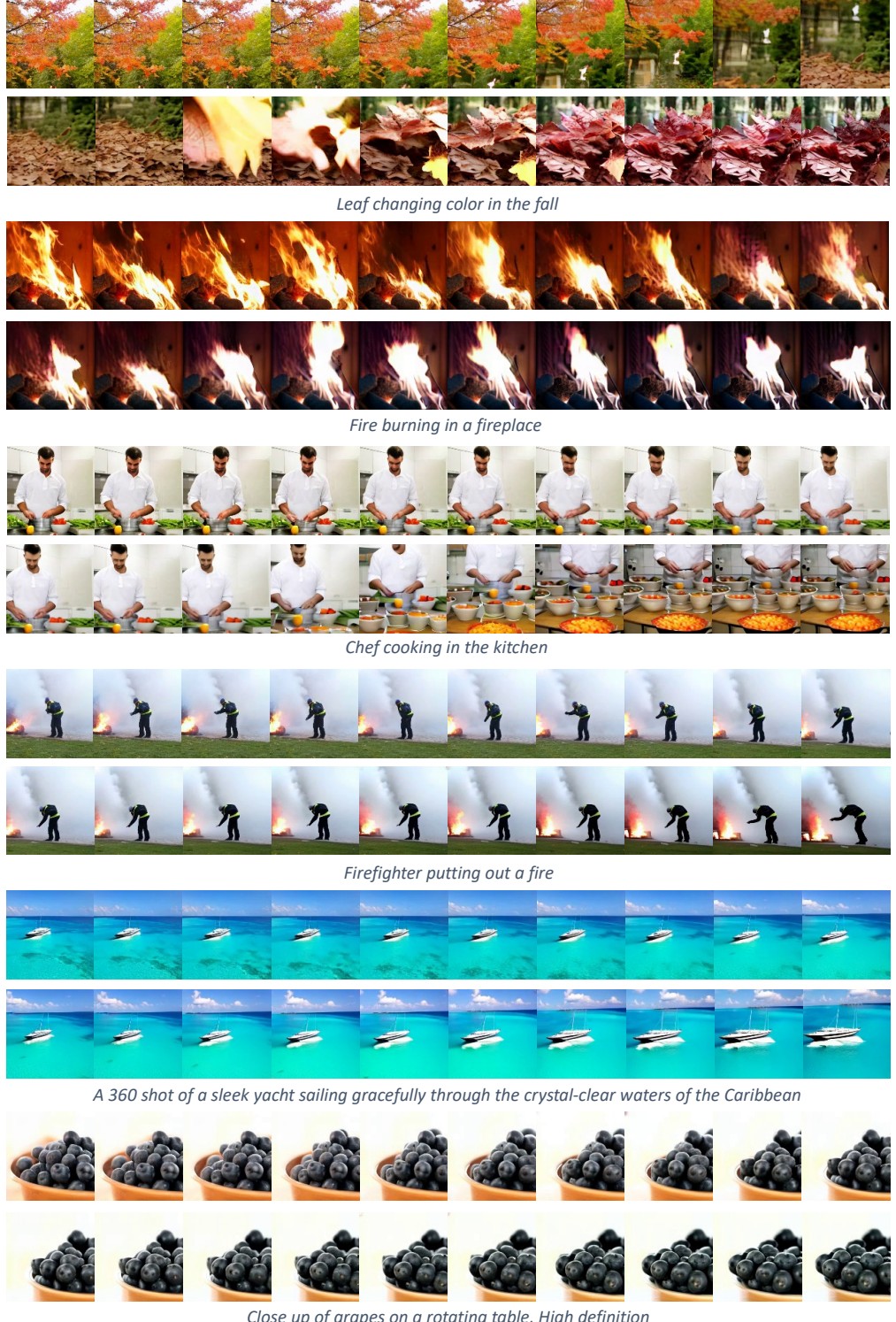

*Leaf changing color in the fall*

*Fire burning in a fireplace*

*Chef cooking in the kitchen*

*Firefighter putting out a fire*

*A 360 shot of a sleek yacht sailing gracefully through the crystal-clear waters of the Caribbean*

*Close up of grapes on a rotating table, High definition*

Figure 16: **Text-to-video generation results** from MALT. We visualize video frames with a stride of 5. We visualize 100 frames in total.

## H    Limitations and Negative Social Impact

While MALT shows strong performance in long video generation benchmarks, it is as-yet tested only with a relatively small model. Considering the scalability of diffusion transformer architectures in text-to-image (Chen et al., 2024b) or text-to-video generation (Gupta et al., 2023), we expect the performance of MALT to show a similar trend given more data and larger models. Moreover, considering that our text-to-video model is trained on short videos (up to 37 frames), training MALT on large-scale text and long-video paired datasets is a natural next direction. Finally, one can combine our approach with other techniques for sequence data, such as Ruhe et al. (2024), or consider recent techniques in LLMs to further expand the context window and to leverage its usefulness (Liu et al., 2023).

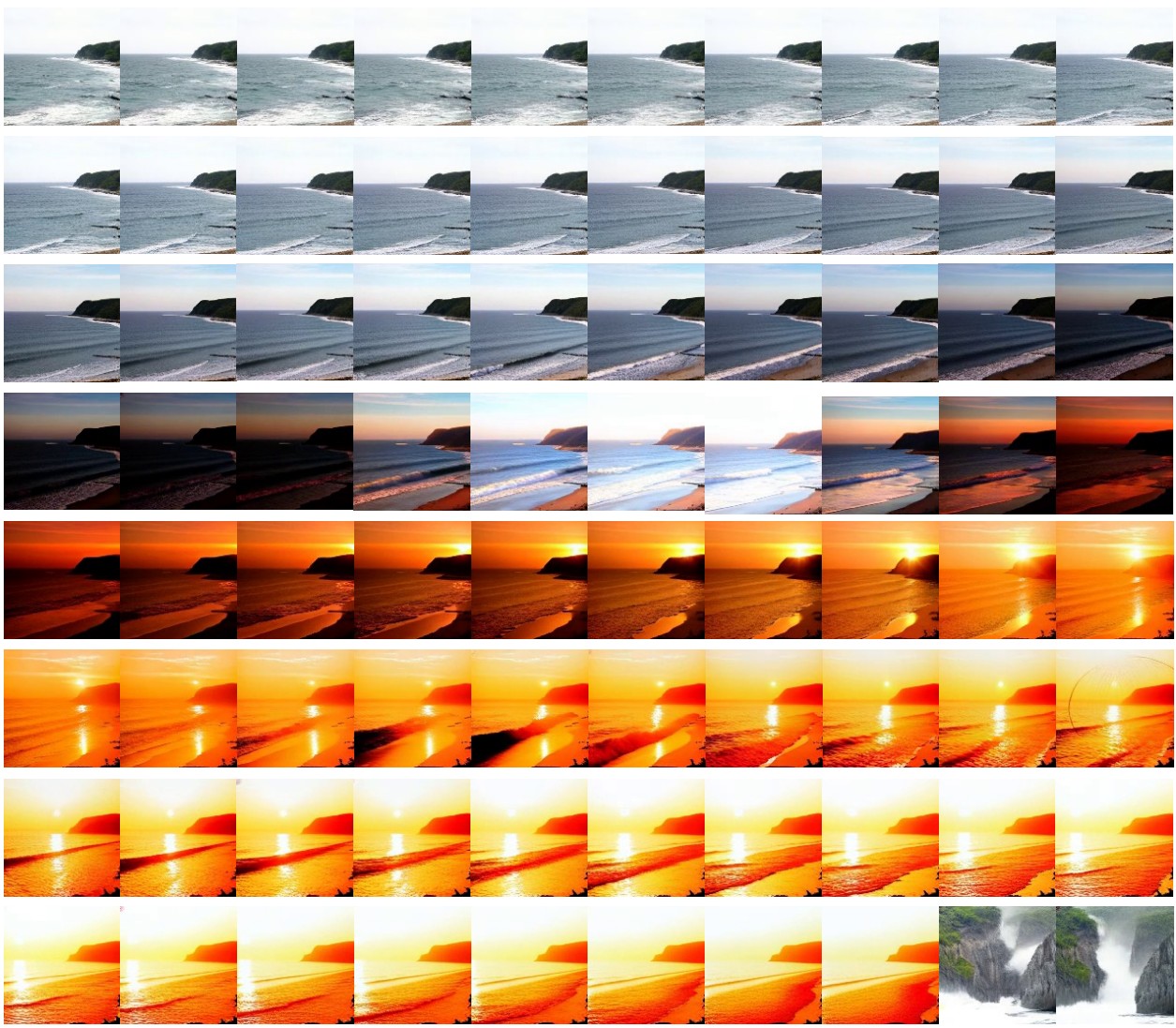

*A tsunami hitting the coast of Japan, in ukiyo-e style.*

Figure 17: **Failure case** from MALT. We visualize the video frames with a stride of 5. Each video frame has a 128×128 resolution. Text prompts used for generating videos are denoted in the below per each visualization. We visualize first 400 frames.

**Failure cases.** For text-to-video generation, our model sometimes does not align with the text prompt: for instance, in Figure 17, the generated video frames do not have "ukiyo-e" style, but they are rather photorealistic. Moreover, while MALT shows strong length generalization longer than the training length (*e.g.*, 50 seconds consistent generation in Figure 6), it sometimes fails to generalize and produce uncorrelated frames to the previous context, as shown in the last two frames in Figure 17. We hypothesize this is because of the small model size and short video length that we used, and we strongly believe training a larger model on long video datasets can solve this issue, considering our superior performance on Kinetics-600 and UCF-101, which are directly trained on long videos.

**Long-context understanding.** While MALT shows superior performance in long video generation with efficiency, there might be better diffusion model architecture and formulation to handle long sequences better with large window sizes, and exploring this should be interesting future work. In particular, one may focus on an approach to model extremely long (*e.g.*, 1, 1-hour movies) with diffusion models, which might be suboptimal with MALT because we always assume the fixed-size memory latent vector. One may consider borrowing ideas in recent LLMs into diffusion model contexts, like we have done in this paper, to apply recurrent mechanisms to diffusion models.

**Negative social impact.** Our problem of interest, video generation, can provide various real-world applications, including content-creations for designers (Villegas et al., 2023), similar to the impact of recent text-to-image diffusion models (Brooks et al., 2023; Meng et al., 2022). At the same moment, one should be aware of the opposite aspect of video generation, as it may unexpectedly result in generating sensitive content (*e.g.*, Deepfake (Guera & Delp, 2018)), as well as generating videos that violate copyright To prevent this, one should first carefully curate video data for training not to include inappropriate contents, and similar to language domain, one should investigate a difference between generated data and real data and develop a detection framework (Mitchell et al., 2023).

