# OpenReview forum: "MALT Diffusion: Memory-Augmented Latent Transformers for Long Video Generation"
_TMLR — Rejected by TMLR_

### Review · Reviewer_jvR9 · 2025-06-20

**Summary Of Contributions:**

This paper introduces MALT (Memory-Augmented Latent Transformers), a novel diffusion model specifically designed for generating long videos, addressing the limitations of existing models that are typically restricted to short clips. MALT handles extended video generation by subdividing videos into short segments and generating them block autoregressively (i.e., sliding window). A key innovation is the use of recurrent attention layers to encode multiple segments into a compact memory latent vector, allowing the model to condition on a long temporal context in the past and continuously generate new footage with consistent quality. The paper also proposes a training technique of including noise augmentation to the memory vector to mitigate error accumulation and ensure long-term stability in generated videos. MALT demonstrates superior performance on long video generation and prediction benchmarks, outperforming previous state-of-the-art methods in terms of contextual understanding and stability, and is shown to produce long text-to-video generations without significant quality degradation.

The effectiveness of MALT was validated through extensive experiments on popular long video benchmarks. For unconditional video generation (of 128 frames), they used the UCF-101 dataset, measuring performance with the Fréchet Video Distance (FVD). For long video prediction, they employed the Kinetics-600 dataset, predicting 80 frames conditioned on 20 initial frames, and evaluating with FVD, Peak Signal-to-Noise Ratio (PSNR), Structural Similarity Index Measure (SSIM), and Learned Perceptual Image Patch Similarity (LPIPS). For text-to-video generation, a version of MALT was trained on 89 million text-short-video pairs and 970 million text-image pairs. Ablation studies were conducted on Kinetics-600 to analyze the contributions of recurrent memory and noise augmentation during training for long-term stability. Error accumulation over generated segments was analyzed by computing FVD values on segments of generated videos over multiple segments. Baselines included existing long video generation and prediction methods such as MoCoGAN, StyleGAN-V, PVDM, TECO, and Perceiver AR.

**Audience:**

Yes

**Broader Impact Concerns:**

The paper lacks any commentary on potential negative social impact, despite there being a section for it in the appendix. That section just talks about the limitations of their approach. I suggest the authors add commentary on the potential negative social impacts of their research.

**Claims And Evidence:**

Yes

**Requested Changes:**

For requested changes I believe are critical in securing my recommendation, I will prefix them with an asterisk (*). For requested changes I believe are high priority but not critical in securing my recommendation, I will prefix them with a plus symbol (+).

- +Fig 1 (right): I suggest adding a label and a short description that describes the memory modules. The yellow and green circles aren't obvious in their differences, meaning, and relationship with one another.
- Section 2 intro section: "We write a sequence of vectors" -> "We write a sequence of tensors"
- Section 2.1, before eq 3: "is identical for the SDE:" -> "is identical to the reverse SDE:"
- *In section 2.1, I suggest providing the equations for sigma(t) and sigma^dot(t).
- *In section 2.1, the DSM objective shows x_0 + eps, but shouldn't eps be scaled by the noise schedule?
- +In section 2.1, I suggest showing which distributions x, eps, and t are sampled from in the DSM objective equations (both the non-latent and latent versions).
- *Eq 6: Why isn't the stop grad applied to the entire hidden state h^i instead of just to its hidden state input h^{i-1}? The same question applies to eq. 8.
- *In the "Training for long term stability" paragraph and in eq. 8, shouldn't z^n be z_0^n?
- *Eq 8: Should h^{n-1} be \tilde{h}^{n-1}?
- *Eq 9: Should h^0 be \tilde{h}^0?
- *Eq 10: I suggest adding how t was sampled (I'm guessing uniformly).
- +Eq 10: I suggest providing the weighting function used.
- *Eq 10: I suggest defining the equation used to obtain z_t^{n+1}, i.e., how the forward SDE is applied.
- *Fig 3 and elsewhere: I suggest describing what stride means in the context of block autoregressive processing.
- *Fig 3: The third row of results in Fig 3 was talked about in section 4.2 of the main manuscript, which is great, but I suggest also providing some brief commentary on the first two rows of results as well. Especially with the second row, it's difficult to tell gauge the quality of the results, so having some commentary like what was provided for the third row would be helpful.
- *Fig 4: It would be nice to provide some brief commentary of the results for the makeup brush video.
- *Fig 6: FIFO-diffusion, StreamingT2V, and FreeNoise aren't mentioned in the baselines paragraph in the main manuscript.
- *Section 4.3, second paragraph: "T2V" -> "text-to-video (T2V)"
- *Section 4.3, second paragraph: I suggest talking a bit about how your model compares to baselines for T2V.
- Section 4.4 name: "System-level comparison" isn't very clear. Particularly the "system-level" part. Perhaps a better section name would be "Quantitative comparison? Up to you though, this is low priority.
- Table 1 caption: "generation and models" -> "generation with various models"
- *Table 1: Latte, FreeNoise, MeBT, CoordTok, and SiT-L/2 aren't mentioned in the baselines paragraph in the main manuscript. It's awkward to see these models pop up in this table and not be talked about anywhere in the main manuscript (not including the appendix).
- Table 2: You forgot a space before "Last only".
- Table 2: "only on the last segment" -> "only on the previous segment"
- *Table 2 caption: The "Robust" column isn't talked about in the caption. You should briefly explain what it represents.
- *Table 2: I suggest bolding first-place results.
- +Table 2: I suggest highlighting which model is the final MALT model (e.g., providing "(ours)" next to "Recurrent").
- *Table 2: If W.A.L.T.-L is the baseline MALT model with all MALT features removed, I would clarify this in the caption.
- Section 4.5, memory design paragraph: "uses the last segment" -> "uses the previous segment"
- *Appendix, algorithm 1: Correct me if I'm wrong, but shouldn't "i in {0, ..., M-1}" be "t in {T, ..., 1}"? If this is true, then don't forget to change "z_M^n" to "z_0^n" in line 7 and the equations in lines 4 and 5.
- *Appendix: I strongly suggest providing an algorithm for the training procedure, especially since the paper focuses on a modification of the DSM training procedure and there's the complication of training with the hidden state mechanism.
- *No discussion of social impact found in either the main manuscript or appendix. I suggest adding a paragraph in the appendix briefly stating potential negative social impacts.

**Strengths And Weaknesses:**

Strengths
- Quality
    - Most claims are well supported with experimental results, both quantitative and qualitative.
    - The methods used to achieve long-term contextual understanding (via recurrent memory) and long term stability (via noise augmentation of hidden state input during training) are sensible.
    - The authors are honest and careful about evaluating both the strengths and weaknesses of their work. I was particularly fond of the honesty behind their analysis of the limitations of MALT in Appendix H.
    - The submission is technically sound for the most part, aside from some clarity issues regarding variables used in equations and some missing information related to tables and figures. These are detailed in the requested changes.
- Clarity
    - The language and narrative flow of the paper is clear and concise for the most part, especially how the authors stated their goals, motivation, and overall approach in the first couple of pages.
- Significance
    - The results presented are significant—the authors showed, both qualitatively and quantitatively, a large improvement over baseline approaches for both long term video generation and prediction, and long term contextual consistency in video generation.
    - The memory-augmented cross-attention mechanism can easily be slotted into any transformer backbone for enabling long-term block autoregressive diffusion-based generation and/or prediction. I can see researchers likely to build off of this idea.
- Originality
    - Related work is more than adequately cited and in each case the authors clearly situate their work in relation to prior work, commenting on how their approach differs.
    - The presented method is new and can be clearly differentiated from prior work.

Weaknesses
- Quality
    - No glaring issues relating to quality.
- Clarity
    - There are some issues relating to some missing and incomplete information in tables and figures, but nothing egregious and these are easily fixable. Please take a look at my requested changes.
    - Some mathematical notation hiccups, but nothing egregious. These are also easily addressable—take a look at my requested changes.
    - Missing commentary on potential negative social impact.
- Significance
    - N/A
- Originality
    - N/A

Overall, there are no glaring and egregious weaknesses of the paper. There are some issues relating to clarity, but I believe these are easily addressable, as shown in my requested changes. The paper is clearly written for the most part and provides a clear and significant contribution with both qualitative and quantitative results to back it up. I can say that at least for me, personally, I would benefit from the techniques introduced in this paper and can see many of my colleagues benefitting the same.

---

> ### Author Response · Authors · 2025-07-15
> **Response to Reviewer jvR9 (1/2)**
>
> We deeply appreciate your insightful comments and efforts in reviewing our manuscript. We respond to each of your comments one by one in what follows. In the revised draft, we mark our major revisions as “blue”.
>
> ---
>
> **[R1] Fig 1 (right) Adding a label and a short description that describes the memory module**
>
> Following your suggestion, we updated Figure 1 by adding a label and a short description that describes the memory module.
>
> ---
>
> **[R2] We write a sequence of vectors" -> "We write a sequence of tensors"**
>
> Thanks for the suggestion. We updated it in the revision.
>
> ---
>
> **[R3]  "is identical for the SDE:" -> "is identical to the reverse SDE:"**
>
> Thanks for pointing it out, and we updated it in the revision.
>
> ---
>
> **[R4] (Section 2.1) Providing the equations for $\sigma(t) and \sigma(t)$.**
>
> Thanks for pointing it out. We provided explicit equations in the revision.
>
> ---
>
> **[R5] (Section 2.1) Noise scheduling in the DSM objective.**
>
> It includes the noise schedule since $\epsilon$ is sampled from $\mathcal{N}(0, \sigma^2I)$. We clarifed it in the revision.
>
> ---
>
> **[R6] showing which distributions x, eps, and t are sampled from in the DSM objective equations (both the non-latent and latent versions)**
>
> Thanks for the suggestion. Each $\mathbf{x}$ is sampled from data distbribution, $\epsilon$ is sampled from $\mathcal{N}(0, \sigma^2I)$, and $t$ is uniformly sampled from $[0, T]$. For latent version we encode each $\mathbf{x}$ into $\mathbf{z}$.
>
> ---
>
> **[R7] (Eq. 6) Stop-grad operation to the entire hidden state instead of  h^i instead of just to its hidden state input h^{i-1}?**
>
> This is because to optimize the diffusion model for both an encoder to encode previous segments and a generator to generate conditioned on previous segments. Similar optimization is applied in previous Transformer-XL (in the language domain) too.
>
> ---
>
> **[R8, 9] (Eq. 8) Should $h^{n-1}$ be $\tilde{h}^{n-1}$? And (Eq. 9)  Should h^0 be \tilde{h}^0?**
>
> $h^{n-1}$ is correct because we add a small noise only at the computation of $\tilde{h}^{n}$ from $h^{n-1}$, indicating the
> computation from  $h^0$ to $h^{n-1}$ remains the same.
>
> ---
>
> **[R10]: (Eq. 10) Sampling distribution $p(t)$.**
>
> Thanks for pointing it out. $t$ is sampled uniformly, and we added it in the revision.
>
> ---
>
> **[R11]  Providing the weighting function used.**
>
> We used $\lambda(t)=1$ in our experiments. We added it in the revision.
>
> ---
>
> **[R12] (Eq. 10) Defining the equation used to obtain z_t^{n+1}, i.e., how the forward SDE is applied.**
>
> Thanks for pointing it out. We added it as Eq. 8 in the revision.
>
> ---
>
> **[R13] Fig 3 and elsewhere: I suggest describing what stride means in the context of block autoregressive processing.**
>
> We clarified in the revision that X strides means that we provided every X frames as a visualization.
>
> ---
>
> **[R14] (Fig. 3, Section 4.2) provides some brief commentary on the first two rows of results as well. Especially with the second row, it's difficult to tell**
>
> Thanks for the suggestion. We added the following description in the revision: Moreover, the videos in the first and second rows depict different actions and motions across the frames, rather than just static or repetitive patterns.
>
> ---
>
> **[R15] Fig 6: FIFO-diffusion, StreamingT2V, and FreeNoise aren't mentioned in the baselines paragraph in the main manuscript.**
>
> Thanks for pointing it out. We added the baseline in the baselines paragraph in the revision.
>
> ---
>
> **[R16] T2V -> text-to-video (T2V)**
>
> Thank you, and we updated it in the revision.
>
> ---
>
> **[R17] Section 4.3, second paragraph: talking a bit about how your model compares to baselines for T2V.**
>
> Thanks for your constructive comment. We added the following sentence in the revision: other baselines fail to achieve this, e.g., either the wine is not consistently poured, or the wine glasses are not consistently maintained.
>
> ---
>
> **[R18] Section 4.4 name: "System-level comparison" isn't very clear.**
>
> Following your suggestion, we updated it to “Qualitative results”  in the revision.
>
> ---
>
> **[R19] Table 1 caption: "generation and models" -> "generation with various models"**
>
> Thanks for finding this, and we updated it in the revision.
>
> ---
>
> **[R20] Table 1: Latte, FreeNoise, MeBT, CoordTok, and SiT-L/2 aren't mentioned in the baselines paragraph in the main manuscript.**
>
> Thank you for your careful review. We have added those baselines in the baselines paragraph in the revision.
>
> ---
>
> **[R21/22/27] Table 2: A space before "Last only" / Table 2, Section 4.5, memory design paragraph: "uses the last segment" -> "uses the previous segment"**
>
> Thanks a lot for careful reading. We updated it in the revision.
>
> ---
>
> **[R23] Table 2: The "Robust" column isn't talked about in the caption.**
>
> Thanks for pointing it out. We explained it in the caption of Table 2 in the revision.

---

> ### Author Response · Authors · 2025-07-15
> **Response to Reviewer jvR9 (2/2)**
>
> **[R24] Table 2: bolding first-place results.**
>
> We now mark the first-place results in bold.
>
> ---
>
> **[R25] Table 2: Highlighting which model is the final MALT model (e.g., providing "(ours)" next to "Recurrent").**
>
> Thanks for your constructive comment, we have now added “ours” next to “Recurrent” in the revision.
>
> ---
>
> **[R26] Table 2: W.A.L.T.-L is the baseline MALT model with all MALT features removed?**
>
> You are correct, and thanks for the suggestion. We have now clarified it in the revision,
>
> ---
>
> **[R28] Appendix, Algorithm 1: Correct me if I'm wrong, but shouldn't "i in {0, ..., M-1}" be "t in {T, ..., 1}"?**
>
> Thanks for pointing this out. We update the typos accordingly in the revision.
>
> ---
>
> **[R29] Appendix: Providing an algorithm for the training procedure.**
>
> Following your suggestion, we added an algorithm for the training procedure in Appendix A.
>
> ---
>
> **[R30] Appendix: Potential negative social impacts.**
>
> Following your suggestion, we added negative social impact in Appendix H.

---

### Review · Reviewer_UNLF · 2025-06-22

**Summary Of Contributions:**

This paper presents MALT Diffusion, a diffusion-based model tailored for long video generation, with two key technical contributions. First, it introduces **recurrent attention layers** that recursively encode multiple video segments into a compact memory latent vector, enabling the model to maintain long-term temporal context and perform segment-level autoregressive generation. Second, it proposes a training technique, i.e., noise augmentation, which significantly improves the stability and consistency of frame quality over extended time horizons. Together, these innovations enable MALT to generate high-quality long-form videos beyond the limitations of prior methods.

**Audience:**

Yes

**Claims And Evidence:**

No

**Requested Changes:**

See Weakness.

**Strengths And Weaknesses:**

Strengths:
1. The paper is well-motivated and clearly written, making it accessible and easy to follow.
2. The proposed method achieves competitive results on multiple benchmarks, indicating its practical value in both video generation and prediction settings.
3. The autoregressive generation framework based on latent diffusion transformers is a compelling design choice that stands out and contributes to effective long-range modeling.

Weaknesses:
1. Technical contribution needs clarification. The paper claims to propose “several training techniques” to improve long-horizon generation, but only noise augmentation is mentioned in detail. If there are indeed multiple distinct techniques, they should be explicitly described and supported with corresponding ablation studies to demonstrate their individual effects.
2. The Method section mostly reads as a pipeline description and lacks clear technical novelty articulation. It would greatly benefit from explicitly stating what aspects of the model architecture or learning strategy are novel compared to prior works, especially in relation to recent latent diffusion or memory-based video models.
3. Training and inference efficiency are not discussed. Since autoregressive generation is typically slower than one-shot methods, and the paper emphasizes the efficiency of memory-augmented attention, it would be helpful to include quantitative comparisons of inference speed and resource usage to substantiate this claim.
4. In Figure 4, the second example does not convincingly demonstrate long-term temporal understanding. In the ground truth, the camel is clearly walking forward, while in the prediction, it appears to move incorrectly. I suggest replacing this example with one that more clearly highlights the model’s ability to capture temporal consistency.
5. Minor issue: there is a typo in the text referencing Equation 7 — it should be corrected from “Eq. equation 7” to simply “Eq. 7.”

---

> ### Author Response · Authors · 2025-07-15
> **Response to Reviewer UNLF**
>
> We deeply appreciate your insightful comments and efforts in reviewing our manuscript. We respond to each of your comments one by one in what follows. In the revised draft, we mark our major revisions as “blue”.
>
> ---
>
> **[W1] Technical contribution needs clarification.**
>
> In addition to the noise augmentation technique, we adopt a stop-grad operation when computing memory during model training. We find that this strategy allows the model to behave both as an encoder that encodes previous contexts and as a generator that generates a video without incurring extreme memory costs. Without this technique, memory consumption increases very rapidly, and the model cannot be trained under limited memory constraints.
>
> ---
>
> **[W2] What aspects of the model architecture or learning strategy are novel compared to prior works? Especially in relation to recent latent diffusion or memory-based video models**
>
> A main novel architecture is the compact memory latent vector based on the recurrent computation of hidden states. By doing so, MALT can generate long videos with longer context lengths compared to recent latent diffusion models, and can encode long videos as compact latent vectors in the hidden states of diffusion transformers.
>
> ---
>
> **[W3] Training and inference efficiency**
>
> Thanks for pointing that out. The additional computational overhead of our memory attention is negligible. Specifically, compared with the vanilla WALT model without memory attention, MALT increases the computation/GFLOPs by about 1.8% to generate a single video segment of 17 frames, thanks to the factorized attention design used for memory attention. Also note that this gap becomes larger as the target video length increases, because the computational cost of MALT scales linearly with the target length due to the segment-wise autoregressive generation, whereas WALT scales quadratically due to the spatiotemporal window attention. As for the number of parameters, MALT-L has 545M parameters, whereas the vanilla WALT-L has 419M parameters. Finally, in terms of memory consumption, MALT requires a fixed amount of memory, while naive DiT memory requirements grow with the video length and thus have a maximum length they can process. In particular, we observe that the vanilla WALT-L model cannot be trained on video clips longer than 57 (= 15 latent frames), whereas MALT can be trained on minute-level videos.
>
> ---
>
> **[W4] The Second example in Figure 4 does not convincingly demonstrate long-term temporal coherence.**
>
> As mentioned in Section 4.2, we provided a camel video as an example because there is an orange roof on the left side of the given prefix frames, and the roof is covered by the camel's head for a few dozen frames before reappearing in the last predictions. Without a long understanding of the context, the model cannot remember these details, and the final predicted segment cannot contain this roof. Nevertheless, we agree with your concern and added a new video in the revision.
>
> ---
>
> **[W5] Typo in Equation 7,**
>
> Thanks for pointing it out. We updated it in the revision.

---

### Review · Reviewer_gtor · 2025-07-01

**Summary Of Contributions:**

This paper addresses the problem of long video generation, a challenging task because it requires coherent, contextually relevant content over extended durations. The authors identify two core obstacles: long-term contextual understanding and long-term stability. To tackle both, they propose a memory-augmented Transformer that uses recurrent attention layers to encode multiple segments into a compact latent memory vector. Additionally, they introduce a novel training strategy and a tailored model-optimization scheme to mitigate inference-time error accumulation and to contain the rapid growth of computational and memory costs as video length increases. Their approach is evaluated on the UCF-101 and Kinetics-400 datasets.

**Audience:**

Yes

**Broader Impact Concerns:**

The authors have provided a Broader Impact Statement, which discusses the potential risks and benefits of their work in the Appendix.

**Claims And Evidence:**

No

**Requested Changes:**

Please see the Weaknesses section for detailed comments.
I suggest the authors address these issues to strengthen their paper.

**Strengths And Weaknesses:**

# Strengths

1. **Clarity and Reproducibility**
   The writing is clear and concise, making the proposed method and its significance easy to grasp. Additionally, the authors provide extensive implementation details, which greatly enhance the reproducibility of their work.

2. **Well-Motivated and Thoroughly Evaluated**
   The motivation for this research is clearly articulated, and the proposed method is rigorously justified. Experiments on benchmark datasets demonstrate the approach’s effectiveness.

# Weaknesses

1. **Inconsistencies between Figures and Equations**
   - In Figure 1, the model is depicted as accessing all previous segments’ hidden states during inference. However, Equations 8–10 show that only the last segment’s hidden state is actually used.
   - In the left panel of Figure 2, only \(h_d^{n-1}\) is shown as key and value, yet Section 2.3 describes concatenating \(h_d^{n-1}\) and \(h_d^{n}\) to form both key and value.

2. **Overstated Claims about Video Length**
   The title and introduction claim the method can generate “arbitrarily long” videos, but quantitative experiments are limited to videos of up to 80 or 128 frames. Although qualitative examples reach 1,300 frames, they lack quantitative evaluation. I recommend replacing “arbitrarily long” with “long” or “extended,” and adding quantitative results on longer sequences (e.g., 512 or 1,024 frames) to substantiate the claim.

3. **Missing Efficiency Metrics**
   As a video-generation method, it is important to report GFLOPs, inference time, number of parameters, and memory usage. Including these metrics would allow readers to assess the method’s computational efficiency, especially for long video synthesis.

4. **Unfair Stability Comparison**
   The authors use the difference between the first and last FVD values to assess stability, but there is a large performance gap between TECO and the proposed method (225.5 vs. 174.3). This disparity suggests the comparison may not fairly validate the authors’ stability claims.

5. **Limited Benchmark Coverage**
   Several established video-generation benchmarks—such as VBench, VBench-2.0, and T2V-CompBench—are omitted. Incorporating these into the evaluation would provide a more comprehensive comparison with existing methods.

6. **Marginal Improvement over Baseline**
   In Table 2, the FVD improvement over “W.A.L.T-L + Last only” is modest (423 vs. 392), raising questions about the practical benefit of the proposed enhancements. The authors should provide deeper analysis to justify the method’s added value.

---

> ### Author Response · Authors · 2025-07-15
> **Response to Reviewer gtor (1/2)**
>
> We deeply appreciate your insightful comments and efforts in reviewing our manuscript. We respond to each of your comments one by one in what follows. In the revised draft, we mark our major revisions as “blue”.
>
> ---
>
> **[W1-1] Inconsistencies between Figure and Equations (Figure 1)**
>
> As mentioned in the caption in Figure 1, the red line denotes the “increased receptive field” of the MALT relative to a single segment. Due to the recurrent mechanism of the hidden states used in memory attention, the hidden states encode all previous segments. Thus, unlike a naive conditioning scheme on the last segment, our scheme can be conditioned on all previous segments.
>
> ---
>
> **[W1-2] Inconsistencies between Figure and Equations (Figure 2)**
>
> Thanks for pointing it out. It should be $[h_d^{n-1}, h_d^{n}]$, and we revised Figure 2 accordingly in the revision.
>
> ---
>
> **[W2] Overstated claims about video length**
>
> Thanks for the suggestion. We updated the term “arbitrarily long” to “long” in the revision. For quantitative evaluation, to the best of our knowledge, there is no reliable metric for long video generation, and thus, previous works have also relied on qualitative evaluation (e.g., [1, 2]). Investigating quantitative evaluation for long video generation would be an interesting area for future work. Moreover, note that we have provided 128-frame video generation on UCF-101 and 100-frame video prediction on Kinetics, which have been used as long video generation benchmarks in previous works [3, 4].
>
> [1] FIFO-Diffusion: Generating Infinite Videos from Text without 101 Training, NeurIPS 2024
> [2] FreeNoise: Tuning-Free Longer Video Diffusion via Noise Rescheduling, ICLR 2024
> [3] Rolling Diffusion Models, ICML 2024
> [4] Temporally Consistent Transformers for Video Generation, ICML 2023
>
> ---
>
> **[W3] Missing efficiency metrics**
>
> Thanks for pointing that out. The additional computational overhead of our memory attention is negligible. Specifically, compared with the vanilla WALT model without memory attention, MALT increases the computation/GFLOPs by about 1.8% to generate a single video segment of 17 frames, thanks to the factorized attention design used for memory attention. Also note that this gap becomes larger as the target video length increases, because the computational cost of MALT scales linearly with the target length due to the segment-wise autoregressive generation, whereas WALT scales quadratically due to the spatiotemporal window attention. As for the number of parameters, MALT-L has 545M parameters, whereas the vanilla WALT-L has 419M parameters. Finally, in terms of memory consumption, MALT requires a fixed amount of memory, while naive DiT memory requirements grow with the video length and thus have a maximum length they can process. In particular, we observe that the vanilla WALT-L model cannot be trained on video clips longer than 57 (= 15 latent frames), whereas MALT can be trained on minute-level videos.
>
> ---
>
> **[W4] Unfair stability comparison**
>
> This seems to be a misunderstanding because smaller FVD values indicate better results, and we compare the FVD **gap** to measure long-term stability. In particular, as mentioned in Section 4.3, the FVD difference between the first and last FVD values of MALT is smaller (174.3) than that of TECO (225.5), even though the FVD difference of MALT is measured with 2$\times$ more segments (i.e., MALT generates 2$\times$ more video frames). This indicates that the frame quality of MALT drops more slowly than that of TECO, highlighting the robustness of MALT to error accumulation.
>
> ---
>
> **[W5] Limited benchmark coverage**
>
> The intent of our work is not to go head-to-head against the latest T2V models, which, among other things, would require a significantly larger model size, a significantly higher dataset scale/quality (c.f. Footnote 2), and training at higher resolutions. These are all issues orthogonal to long-term stability, which we isolate as the focus of our submission. Moreover, even if we were to train our long-term model in such a scaled-up regime, current evaluation metrics like VBench focus on relatively short clip fidelity and have not been proven to be reliable metrics in a long video regime. Thus, several long video generation approaches [1] have provided only qualitative comparisons on T2V and have conducted their quantitative evaluation on popular long video benchmarks (e.g., UCF-101) --- we have followed this approach. We believe that our T2V results are a good proof of concept to show what might be possible, but we are happy to remove them if the reviewers insist, as we believe the benefits of our method are adequately shown in the UCF-101 and Kinetics experiments and are done at scales that are reproducible in academic labs.
>
> [1] FIFO-Diffusion: Generating Infinite Videos from Text without Training, NeurIPS 2024

---

> ### Author Response · Authors · 2025-07-15
> **Response to Reviewer gtor (2/2)**
>
> **[W6] Marginal improvement over baseline**
>
> First, note that [W.A.L.T++ Last only] applies our noise-adding technique to ensure long-term stability (which is one of our technical components). This noise-adding technique reduces the train–inference gap during training, enabling the model to generate video frames with consistent quality, and thereby improves the FVD score compared with the vanilla W.A.L.T, which is our baseline (although it still lacks long-term contextual understanding capability due to the limited context length).
>
> To validate the enhancements in long-term contextual understanding capability through our recurrent memory design, recall that we also provided a qualitative example in Figure 4. Specifically, there is an orange roof on the left side of the given prefix frames, which is covered by the camel's head for a few dozen frames before reappearing in the final predictions. Without a long-term understanding of the context, the model cannot remember these details, and the final predicted segment does not contain this roof.

---

### Decision · Action_Editor_6EWH · 2025-08-12

**Recommendation:** Reject

**Additional Comments:**

The reviewer feedback is mixed, with both positive recognition of the topic’s relevance and concerns about the strength of the evidence supporting the claims. The main issues raised include insufficient clarity and substantiation of the technical contributions, particularly the stop-grad operation; a lack of explicit articulation of novelty compared to recent latent diffusion and memory-based video models; and efficiency claims that are not backed by comprehensive experiments on speed, memory usage, and scalability. The authors are encouraged to address these issues thoroughly and resubmit after substantial revision.

**Audience:**

Yes

**Audience Explanation:**

The topic would likely interest parts of TMLR’s audience, especially those working on generative video models. However, stronger and more comprehensive experimental evidence would make the findings more convincing and broaden their appeal.

**Claims And Evidence:**

No

**Claims Explanation:**

Overall, the claims in the submission are only partially supported by convincing and clear evidence. While the authors address some reviewer concerns in their response, key contributions such as the stop-grad operation are not sufficiently discussed or evaluated in the main paper. Similarly, the efficiency claims lack direct, thorough experimental support, and the added example for long-term temporal coherence is not accompanied by deeper analysis.

**Resubmission Of Major Revision:**

The authors may consider submitting a major revision at a later time.